# A$^3$E: Towards Compositional Model Editing

**Hongming Piao**
City University of Hong Kong
hpiao6-c@my.cityu.edu.hk

**Hao Wang**
City University of Hong Kong
hao.wang@my.cityu.edu.hk

**Dapeng Oliver Wu**[†]
City University of Hong Kong
dapengwu@cityu.edu.hk

**Ying Wei**[†]
Zhejiang University
ying.wei@zju.edu.cn

## Abstract

Model editing has become a *de-facto* practice to address hallucinations and outdated knowledge of large language models (LLMs). However, existing methods are predominantly evaluated in isolation, i.e., one edit at a time, failing to consider a critical scenario of compositional model editing, where multiple edits must be integrated and jointly utilized to answer real-world multifaceted questions. For instance, in medical domains, if one edit informs LLMs that COVID-19 causes "fever" and another that it causes "loss of taste", a qualified compositional editor should enable LLMs to answer the question "What are the symptoms of COVID-19?" with both "fever" and "loss of taste" (and potentially more). In this work, we define and systematically benchmark this compositional model editing (CME) task, identifying three key undesirable issues that existing methods struggle with: *knowledge loss*, *incorrect preceding* and *knowledge sinking*. To overcome these issues, we propose A$^3$E, a novel compositional editor that (1) *adaptively combines and adaptively regularizes* pre-trained foundation knowledge in LLMs in the stage of edit training and (2) *adaptively merges* multiple edits to better meet compositional needs in the stage of edit composing. Extensive experiments demonstrate that A$^3$E improves the composability by at least 22.45% without sacrificing the performance of non-compositional model editing. The code is available at https://github.com/piaohongming/A3E.

## 1 Introduction

LLMs learn extensive knowledge from massive pre-training corpora and utilize the learned knowledge during inference to meet a wide range of tasks [1, 2, 3, 4]. Despite their impressive capabilities, especially as the pre-training data and model size scale [5], LLMs remain prone to factual hallucinations [6] and outdated knowledge [7]. These errors often emerge gradually after deployment and significantly impairs system reliability, which necessitates timely and effective corrections. Conventional solutions such as re-training are not only time-consuming, but fine-tuning without access to pre-training data also poses a high risk of interfering with irrelevant knowledge. Therefore, model editing [8] has emerged as a promising approach that enables targeted updates to LLMs in a data-efficient manner (minimal reliance on pre-training data), localized scope (limited impact on irrelevant knowledge), and with high reliability (accurate correction of the targeted knowledge).

Efforts toward model editing center around improving reliability [9, 10], generalization [10, 11], and locality [8, 12], with further studies exploring editing ripple effects [13], bidirectional generalization [14], and its impact on downstream tasks [15, 16, 17, 18]. Progress along with these attempts is

---

[†]Corresponding authors.

39th Conference on Neural Information Processing Systems (NeurIPS 2025).

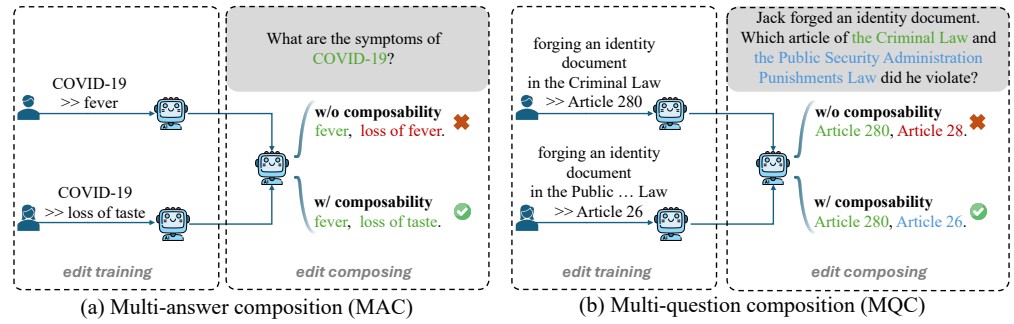

|  | (a) Multi-answer composition (MAC) | (b) Multi-question composition (MQC) |
|---|---|---|

Figure 1: There are two stages in CME: *edit training* and *edit composing*. (a) An example for MAC: Multiple symptoms for a disease. (b) An example for MQC: Multiple laws involved in one crime.

unanimously evaluated under a *single-edit assumption*; that is, LLMs recall each atomic edit in isolation. For example, if an edit previously taught an LLM that COVID-19 causes "loss of taste", evaluations only test whether this LLM can correctly recall this isolated fact, e.g., "I have loss of taste, do I have COVID-19?" Unfortunately, this evaluation paradigm diverges significantly from real-world demands. In many practical domains, such as medicine and law (as shown in Fig. 1), questions likely require access to multiple edits simultaneously. In Fig. 1(a), for instance, answering a multi-symptom diagnostic question requires composing two pieces of knowledge (multi-answer composition, MAC); if these facts were introduced through prior edits, LLMs must effectively leverage both. Similarly, in Fig. 1(b), questions may require reasoning over multiple pieces of edited knowledge to form a unified answer (multi-question composition, MQC). In this paper, we formally define and benchmark tasks that demand such compositional capability as Compositional Model Editing (CME) tasks. The success of CME hinges on two criteria: (1) in the *edit training* stage, individual edits must be performed in a way that guards their composability for downstream inference; (2) in the *edit composing* stage, question-related edits must be effectively merged.

As shown in Fig. 2, state-of-the-art editing methods suffer from *knowledge loss*, *incorrect preceding* and *knowledge sinking*, which comprehensively encompass all the failure cases we observed. MEND [9] and ROME [8] modify the entire matrix in the feed-forward netwok (FFN) through hypernetworks and closed-form solutions, respectively, which yields significant *knowledge loss* when composing different edits. AlphaEdit [19] exhibits a similar level of *knowledge loss*, likely because it prevents interference between edits by projecting each update into the null space of unrelated questions. However, in the MAC and MQC settings we target, where a single question depends on multiple edits, this orthogonality collapses and conflicts between edits still persist. WISE [20] alleviates *knowledge loss* by randomly masking parameter updates of each individual edit, though the likelihood of mask overlap grows significantly as the number of edits increases and thus composability degrades. GRACE [21], MELO [22], and T-patcher [23] replace hidden states or insert new key-value pairs in the FFN layer in a low-rank form. Such low storage overhead for each edit enables them to treat all edits as an external vector database attached to the original parameters. Despite reduced *knowledge loss*, all three methods suffer from the pronounced issue of *incorrect preceding*. Collectively, these limitations reveal a pressing gap, i.e., *how to train and compose different edits to improve edit composability?*

Our response to the question is the proposed $\mathbf{A}^3\mathbf{E}$. Specifically, previous works [24, 25] have pointed out that the FFN layer in Transformer [26] is a key-value neural memory [27], where the down projection matrix stores robust and generalizable foundation knowledge pre-trained on large datasets. The composition of these foundation knowledge pieces forms multiple advanced knowledge needed in the generation process of LLMs. We hypothesize and empirically verify that leveraging the pre-trained foundation knowledge in the down projection matrix is sufficient for not only composing the new knowledge but also boosting composability. Thus, in order to *train edits with better composability*, we **A**daptively combine the pre-trained foundation knowledge in the down projection matrix to reduce *knowledge loss*. By **A**daptively regularizing the use of selected knowledge at the last answer token and non-label logits, we effectively alleviate *knowledge sinking* and *incorrect preceding*. In order to *compose edits to further enhance composability*, we utilize a vector database [21, 22] to filter out irrelevant edits in a question-wise way at inference and **A**daptively merge relevant edits with proper combinations of pre-trained foundation knowledge. In summary, our contributions are four-fold.

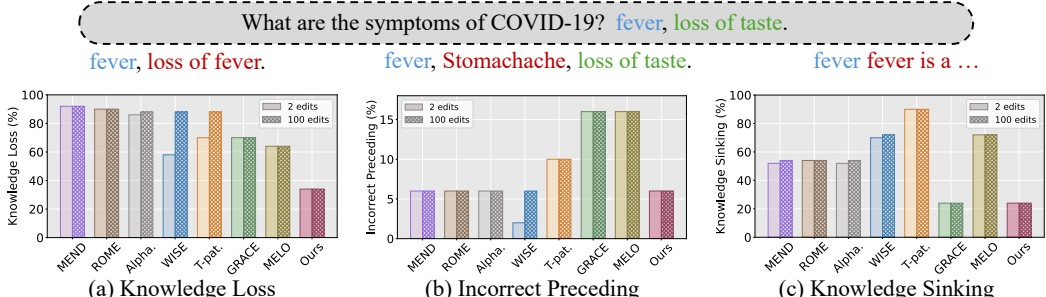

Figure 2: The illustration of *knowledge loss*, *incorrect preceding* and *knowledge sinking*. The text in blue, red, green represents the first correct answer, the wrong answer and the second correct answer. (a) *Knowledge loss:* Part of answers are not correctly generated. (b) *Incorrect preceding:* Wrong answers are generated before the correct answers. (c) *Knowledge sinking:* Context-irrelevant content are generated after prior answers; that is, the starting point for generating subsequent answer "loss of taste" is no longer the question itself (e.g., "What are the symptoms of COVID-19?"), but the question combined with the prior answers (e.g., "What are the symptoms of COVID-19? fever").

- We are the first to define and benchmark the CME evaluation setup, paving the way for future model editors to support multi-edit reasoning that is crucial for real-world use cases.
- We conduct a systematic analysis of existing methods under CME, revealing three pivotal failure modes in composability, including *knowledge loss*, *incorrect preceding* and *knowledge sinking*.
- We develop a dual-stage framework, edit training (adaptive combining and regularizing) and edit composing (adaptive merging), that preserves the effectiveness of individual edits while enabling their synergistic integration during inference.
- Through extensive experiments on two datasets and four CME tasks, we improves the composability by at least 22.45%, without sacrificing the performance of non-compositional model editing.

## 2 Preliminaries

In this section, we first provide a formal definition of compositional model editing, and then demonstrate how to perform edit training and edit composing in FFN. Please refer to Tab. 3 for the definition of notations.

### 2.1 Compositional Model Editing (CME)

In model editing (ME), let $f_\Theta : \mathbb{X} \mapsto \mathbb{Y}$, parameterized by $\Theta$, represents a model mapping the input $x$ to the output $f_\Theta(x)$. Given a model $\Theta_0$ before a single edit $(x_e, y_e)$, let $\mathcal{I}(x_e)$ denote the in-scope input with the same semantics as the edit question. Then the objective of ME is:

$$f_{\Theta_e} = \text{ME}\left(f_{\Theta_0}, x_e, y_e\right), \quad \text{s.t. } f_{\Theta_e}(x) = \begin{cases} y_e & \text{if } x \in \mathcal{I}(x_e), \\ f_{\Theta_0}(x) & \text{if } x \notin \mathcal{I}(x_e). \end{cases} \tag{1}$$

Compositional model editing (CME) contains two stages: *edit training* and *edit composing*. For the *edit training* stage, an edit is trained for $(x_e, y_e)$. For the *edit composing* stage, the model needs to compose the edit for $(x_e, y_e)$ with other edits at inference time to answer a single question. Let $\mathcal{C}(\mathcal{X}_e)$ represent the question that contains the semantics of multiple edit questions $\mathcal{X}_e = \{x_e, x_e^1, x_e^2, ..., x_e^{c-1}\}$ including $x_e$, where $c$ is the composition number. Let $\mathcal{C}(\mathcal{Y}_e)$ represent the target output that contains corresponding multiple answers $\mathcal{Y}_e = \{y_e, y_e^1, y_e^2, ..., y_e^{c-1}\}$ including $y_e$. The shared objective for the two stages becomes

$$f_{\Theta_e} = \text{CME}\left(f_{\Theta_0}, x_e, y_e\right), \quad \text{s.t. } f_{\Theta_e}(x) = \begin{cases} \mathcal{C}(\mathcal{Y}_e) & \text{if } x \in \mathcal{I}(\mathcal{C}(\forall \mathcal{X}_e)), \\ f_{\Theta_0}(x) & \text{else} \end{cases} \tag{2}$$

It is worth noting that, first, the edits involved in edit composing are not necessarily all contained within $(\mathcal{X}_e, \mathcal{Y}_e)$, which introduces interference from edits useless to the current problem. Additionally, within our definition, all edits may appear asynchronously over time and be distributed spatially. Therefore, our definition is applicable to both lifelong model editing and federated model editing.

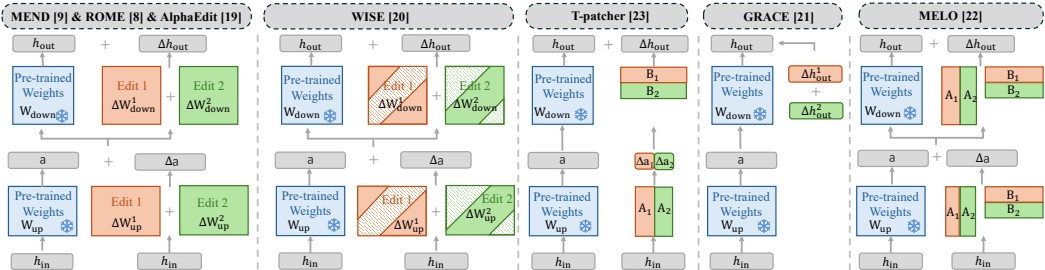

Figure 3: The visualization of different baselines, where MEND, ROME, AlphaEdit and WISE edit either $\mathbf{W}_{\text{down}}$ or both $\mathbf{W}_{\text{up}}$ and $\mathbf{W}_{\text{down}}$ with or without mask. In contrast, GRACE, T-patcher, and MELO edit the $\mathbf{W}_{\text{down}}$ or both $\mathbf{W}_{\text{up}}$ and $\mathbf{W}_{\text{down}}$ in different low-rank forms.

## 2.2 *Edit Training* and *Edit Composing* in FFN

Based on the results in Fig. 2, we categorize the factors influencing the composability of edits into **Composable** and **Context-preserving**. Whether the edits are composable refers to whether there is significant interference between the trained edits, which impacts *knowledge loss* and *incorrect preceding*. Context-preserving, on the other hand, refers to whether the edits affect the model's ability to follow the context to generate other answers, thereby influencing *knowledge sinking*. In this section, we discuss whether *edit training* stage of existing methods effectively have these properties and whether their *edit composing* preserves these properties as shown in Table 1.

Most existing model editing methods insert new knowledge by modifying the parameters or outputs of the FFN layer during the forward process of different tokens. The FFN operates as key-value neuron memories, where the values (rows) in the down projection layer, store pre-trained foundation knowledge obtained from vast amounts of data [24, 25]. During the forward process, it retrieves values from the down projection matrix $\mathbf{W}_{\text{down}} \in \mathbb{R}^{n \times m}$ by matching the keys in the up projection matrix $\mathbf{W}_{\text{up}} \in \mathbb{R}^{m \times n}$ with the input $h_{\text{in}} \in \mathbb{R}^{1 \times m}$:

$$h_{\text{out}} = \text{FFN}(h_{\text{in}}) = \mathbf{a}\mathbf{W}_{\text{down}} + \mathbf{b}_{\text{down}}, \quad \text{where } \mathbf{a} = \text{Act}(h_{\text{in}}\mathbf{W}_{\text{up}} + \mathbf{b}_{\text{up}}). \tag{3}$$

Here, Act is the activation function (e.g., Relu [28], Gelu [29], and SwiGLU [30]). $\mathbf{a}$ is the vector of activation values and serves as the weights to retrieve different pre-trained foundation knowledge in $\mathbf{W}_{\text{down}}$. $\mathbf{b}_{\text{down}} \in \mathbb{R}^{1 \times m}$ and $\mathbf{b}_{\text{up}} \in \mathbb{R}^{1 \times n}$ are bias vectors.

We summarize *edit training* and *edit composing* in FFN for existing methods in Fig. 3. Among them, GRACE only edits the forward process of the last input token while achieving context-preserving. The other methods modify the forward process of all tokens but lack context-preserving. MEND, ROME, AlphaEdit and WISE modify the entire $\mathbf{W}_{\text{down}}$, or $\mathbf{W}_{\text{up}}$ and $\mathbf{W}_{\text{down}}$ to insert new knowledge in *edit training*, thus different $\Delta\mathbf{W}_{\text{up}}$ and $\Delta\mathbf{W}_{\text{down}}$ are added together to utilize multiple pieces of knowledge in *edit composing*. AlphaEdit only edits $\mathbf{W}_{\text{down}}$ and projects $\Delta\mathbf{W}_{\text{down}}$ onto the null space of the preserved knowledge including other edits be-

Table 1: Summary of model editing methods under composable and context-preserving in *edit training* and if *edit composing* preserves these properties.

| Method | Edit Training | | Edit Composing |
|---|---|---|---|
| | Composable | Context-preserving | |
| MEND | ✗ | ✗ | ✗ |
| ROME | ✗ | ✗ | ✗ |
| AlphaEdit | ✗ | ✗ | ✗ |
| WISE | ✔ | ✗ | ✗ |
| T-patcher | ✗ | ✗ | ✗ |
| GRACE | ✗ | ✔ | ✗ |
| MELO | ✗ | ✗ | ✗ |
| $A^3E$ | ✔ | ✔ | ✔ |

fore applying it to $\mathbf{W}_{\text{down}}$ to avoid interference between edits. However, in the CME settings we focus on, where a single question depends on multiple edits, this orthogonality becomes ineffective and conflicts between edits still persist. WISE learns more composable edits by masking part of $\mathbf{W}_{\text{up}}$ and $\mathbf{W}_{\text{down}}$ in *edit training*, but the mask on the whole matrix with no carefully designed composing algorithm struggles to keep a large number of edits composable in *edit composing*. T-patcher concatenates multiple keys $\mathbf{A} \in \mathbb{R}^{m \times r}$ and corresponding values $\mathbf{B} \in \mathbb{R}^{r \times m}$ to $\mathbf{W}_{\text{up}}$ and $\mathbf{W}_{\text{down}}$ respectively. For GRACE, the output $h_{\text{out}}$ are replaced with the sum of edit vectors $\Delta h_{\text{out}} \in \mathbb{R}^{1 \times m}$ for multiple pieces of knowledge. MELO trains a LoRA [31] $(\mathbf{A}, \mathbf{B})$ for each edit, where $\mathbf{A} \in \mathbb{R}^{n \times r}$ and $\mathbf{B} \in \mathbb{R}^{r \times m}$. The combination of different knowledge is achieved by concatenating $\mathbf{A}$ and $\mathbf{B}$ of different LoRAs, respectively. Thanks to the low storage overhead of their low-rank form, GRACE and MELO employ a vector database to store all edits and filter out useless ones during inference,

which achieves more composable *edit composing* with a large number of edits than others but the results contain more *incorrect preceding* and are far from ideal.

# 3 Rethinking the Composability in Model Editing

This section examines existing model editing methods to explore three key questions: *How to train composable edits? How to train context-preserving edits? How to compose edits to further enhance composability?* Our answers are **Finding 1**, **Finding 2** and **Finding 3**, respectively.

**Experimental settings.** We conducted analytical experiments using the Llama3-8B model [32] on the PEAK-CF dataset [33] with 50 2-edit composition samples, During inference, we generate 30 tokens for each question, which is enough to output all edited answers. We analyze the performance of existing model editing methods with the metric $SR\text{-}S = \frac{1}{|\mathcal{D}_{\text{test}}|}\sum_{t=1}^{|\mathcal{D}_{\text{test}}|}\mathbb{I}(\forall a \in \mathcal{Y}_t, \text{rank}(a) < |\mathcal{Y}_t|)$, which is the rate of question $t$ in test dataset $\mathcal{D}_{\text{test}}$ that all edited answers $\mathcal{Y}_t$ are output before irrelevant content, where $\text{rank}(\cdot)$ represents the rank of an answer in *all answers in the output* and $|\mathcal{Y}_t|$ represents the number of edited answers. Please refer to Appendix H for an example.

**Finding 1.** *The combination of pre-trained foundation knowledge is enough for ME while boosting the composability.* As introduced in Sec. 1, $\mathbf{W}_{\text{down}}$ stores pre-trained foundation knowledge obtained from vast amounts of data. Our hypothesis is that the ability of LLMs to combine different advanced knowledge stems from these well-trained foundation knowledge elements. These elements boosts the composability with a more robust and generalizable knowledge representation from large-scale pre-training. We validate our hypothesis by the average performance of methods editing full matrix and in low-rank form when editing $\mathbf{W}_{\text{up}}$ or $\mathbf{W}_{\text{down}}$ at different layers. As shown in Fig. 4(a) and Fig. 4(b), editing $\mathbf{W}_{\text{up}}$ is sufficient to guarantee the performance compared with editing $\mathbf{W}_{\text{down}}$ as well as editing both $\mathbf{W}_{\text{up}}$ and $\mathbf{W}_{\text{down}}$. This means the pre-trained foundation knowledge in $\mathbf{W}_{\text{down}}$ is sufficient to express new knowledge. The generally higher *SR-S* of editing $\mathbf{W}_{\text{up}}$ in different layers shown in Fig. 4(c) confirms our hypothesis. For a more comprehensive discussion about the sufficiency of editing $\mathbf{W}_{\text{up}}$ for higher *SR-S* and less *knowledge loss*, please refer to Appendix I.

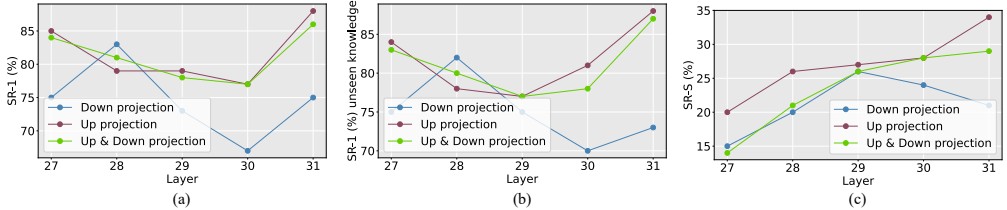

Figure 4: (a) Performance of non-compositional ME measured by *SR-1* when editing $\mathbf{W}_{\text{down}}$, $\mathbf{W}_{\text{up}}$ or both of them. (b) Performance of non-compositional ME measured by *SR-1* when editing $\mathbf{W}_{\text{down}}$, $\mathbf{W}_{\text{up}}$ or both of them with pseudo samples to simulate totally unseen knowledge (Appendix I). (c) Performance of CME measured by *SR-S* when editing $\mathbf{W}_{\text{down}}$, $\mathbf{W}_{\text{up}}$ or both of them.

**Finding 2.** *Less change of the hidden state of the last answer token [la], more context-preserving.* We demonstrate in Fig. 5(a) the relationship between the context-preserving rate and the change of the hidden state at the edited FFN layer when inferring with [la] (e.g., "What are the symptoms of COVID-19? fever"). The results show an evident inverse relationship between these two factors in the baselines, indicating that a smaller change in the hidden state of [la] can achieve better context-preserving. The finding

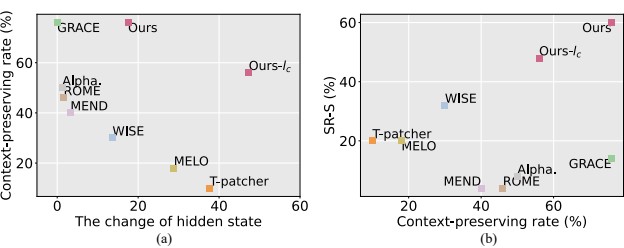

Figure 5: The relevance (a) between the change of hidden state at [la] and the context-preserving rate (b) between context-preserving and *SR-S*.

also answers why GRACE achieves context-following, because it has no parametric interference on the forward process of [la] by editing the last input token only.

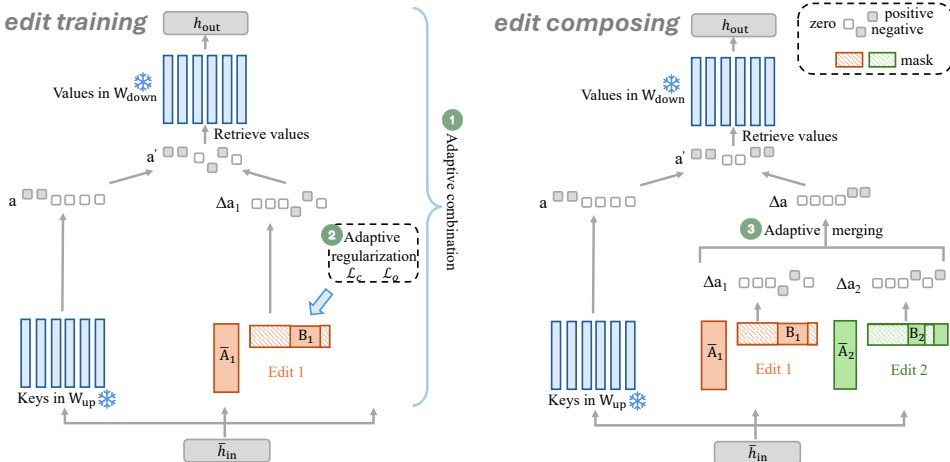

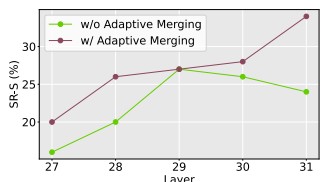

Figure 7: Our proposed A³E, including adaptive combination (Sec. 4.1), adaptive regularization (Sec. 4.1) and adaptive merging (Sec. 4.2).

**Finding 3.** *Setting the elements with different signs in different edits to zero (Adaptive Merging) during edit composing enhances the composability.* The elements with opposite signs between different edits may contain conflicting knowledge. In Fig. 6, inspired by [34], we validate that setting these opposing elements in different edits to zero better balances the knowledge forgetting in the edits and their negative interference on each other. As can be seen, the mean performance of methods editing the full matrix and in low-rank form with *Adaptive Merging* outperforms directly composing different edits.

Figure 6: Performance of CME measured by *SR-S* with or without *Adaptive Merging*.

## 4 The Proposed Method

Although the three findings mentioned above contributes the composability of model editing, there still lacks of a unified framework to fully integrate their advantages. *How to train composable and context-preserving edits?* Based on Finding 1, we adaptively select the pre-trained foundation knowledge using a mask and in a low-rank form to utilize the vector database, namely **Adaptive Combination** of pre-trained foundation knowledge. We propose two **Adaptive Regularization** to regularize the use of pre-trained foundation knowledge inspired by Finding 2 while further alleviating *incorrect preceding* in low-rank form. *How to compose edits to further enhance composability?* We utilize a vector database to filter out useless edits following [21, 22] while proposing **Adaptive Merging** for different combinations of pre-trained foundation knowledge inspired by Finding 3. The method is shown in Fig. 7. Please refer to Appendix D for the vector database and Alg. 1-2 for the complete editing progress.

### 4.1 How to train composable and context-preserving edits?

**Adaptive Combination.** Inspired by Finding 1, we perform model editing only on $\mathbf{W}_{\text{up}}$ in a low-rank form similar to LoRA but incorporate an adaptive mask on $\mathbf{B}$. The adaptive mask aims to only preserve the most related pre-trained foundation knowledge to achieve a better trade-off between the composability and learning ability. Specifically, an $(\mathbf{A}, \mathbf{B})$ pair is trained for each edit following [22]. The forward process of the edited FFN is calculated as:

$$h_{\text{out}} = \text{FFN}_{\text{edit}}(h_{\text{in}}, \mathbf{A}, \mathbf{B}, \mathbf{M}) = \mathbf{a}\mathbf{W}_{\text{down}} + \mathbf{b}_{\text{down}},$$
$$\text{where } \mathbf{a} = \text{Act}(h_{\text{in}}\mathbf{W}_{\text{up}} + \underbrace{\bar{h}_{\text{in}}\bar{\mathbf{A}}(\mathbf{B} \times \mathbf{M})}_{\Delta\mathbf{a}} + \mathbf{b}_{\text{up}}). \tag{4}$$

---

**Algorithm 1** The Edit Training Stage

---
1: **Input** The initial LLM $f_\theta$, the edit dataset $\mathcal{D}_{\text{edit}}$, the initialized vector database $\mathcal{S}_{\text{db}}$.
2: **Output** The vector database $\mathcal{S}_{\text{db}}$
3: **for** each edit $(x_e, y_e) \in \mathcal{D}_{\text{edit}}$ **do**
4:   Compute mask $\mathbf{M}_e$.
5:   Replace $h_{\text{out}} = \text{FFN}(h_{\text{in}})$ (Eq. (3)) with $h_{\text{out}} = \text{FFN}_{\text{edit}}(h_{\text{in}}, \mathbf{A}_e, \mathbf{B}_e, \mathbf{M}_e)$ (Eq. (4)) at selected layers.
6:   Update $(\mathbf{A}_e, \mathbf{B}_e)$ with $\mathcal{L}$ (Eq. (7)).
7:   Save $\mathbf{K}_e$ into $\mathcal{S}_{\text{db}}$ as the key of vector database and save $(\mathbf{A}_e, \mathbf{B}_e)$ as the corresponding value.
8: **end for**

---

$\mathbf{A} \in \mathbb{R}^{m \times r}, \mathbf{B} \in \mathbb{R}^{r \times n}$ are the components of LoRA while $\bar{h}_{\text{in}}$ and $\bar{\mathbf{A}}$ are normalized $h_{\text{in}}$ and $\mathbf{A}$ to avoid the weights of using different $\mathbf{B}$ have significant differences and lead to *knowledge loss*. $\mathbf{M} \in \mathbb{R}^{r \times n}$ is the mask to select top $k$ related values in $\mathbf{W}_{\text{down}}$, where the correlation is calculated by the dot product between values in $\mathbf{W}_{\text{down}}$ and the answer tokens' embedding:

**Adaptive Regularization.** Based on Finding 2, we design a regularization $\mathcal{L}_c$ to restrict the additional use of pre-trained foundation knowledge by the last answer token [la], thereby enabling the context-preserving to generate the next answer. Specifically:

$$\mathcal{L}_c = ||\bar{h}_{\text{in}}^{[\text{la}]} \bar{\mathbf{A}}||_2, \tag{5}$$

where $\bar{h}_{\text{in}}^{[\text{la}]}$ represents the normalized input hidden state of the FFN layer at [la]. $\mathcal{L}_c$ achieves context-preserving by encouraging a low $\Delta\mathbf{a}$ in Eq. (4).

In order to further alleviate *incorrect preceding* in low-rank form, we propose another regularization $\mathcal{L}_o$ to restrict the logits of tokens $h_{\text{logit}}^{[\text{a}]}$ except labels by:

$$\mathcal{L}_o = \sum_{[\text{a}]} \sum_{i \in \mathcal{W}_{[\text{a}]}} [h_{\text{logit}}^{[\text{a}]}]_i, \tag{6}$$

where [a] represents the positions of an edit sample with labels, $\mathcal{W}_{[\text{a}]}$ represents the set of all token IDs except the label of the position [a]. During the *edit training* stage, we optimize $(\mathbf{A}, \mathbf{B})$ with the loss:

$$\mathcal{L} = \mathcal{L}_e + \alpha\mathcal{L}_c + \beta\mathcal{L}_o, \tag{7}$$

where $\mathcal{L}_e$ are cross-entropy loss, $\alpha$ and $\beta$ are hyperparameters.

### 4.2 How to compose edits?

**Adaptive Merging.** During the inference time, after filtering out useless edits by the vector database $\mathcal{S}_{\text{db}}$ (Appendix D), we need to *compose different combinations of pre-trained foundation knowledge* and different edits may conflict in the utilization of pre-trained foundation knowledge. Specifically, some edits aim to leverage the $j$-th value in $\mathbf{W}_{\text{down}}$ to compose the target answer, so $[\Delta\mathbf{a}]_j$ tends to be positive. In contrast, other edits may want to erase the knowledge in the $j$-th row to compose the target answer, so $[\Delta\mathbf{a}]_j$ tends to be negative. Inspired by Finding 3, when such conflicts occur, we set $[\Delta\mathbf{a}]_j$ of all used edits to 0 to avoid the overly strong impact on each other. For simplicity, we represent this merging process of different $\Delta\mathbf{a}$ as $\hat{\sum}$ and the $\Delta\mathbf{a}$ of preserved useful edits by vector database as $\mathcal{S}$. Thus the forward process during inference changes to:

$$h_{\text{out}} = \text{FFN}_{\text{test}}(h_{\text{in}}, \mathcal{S}) = \mathbf{a}\mathbf{W}_{\text{down}} + \mathbf{b}_{\text{down}},$$
$$\text{where } \mathbf{a} = \text{Act}(h_{\text{in}}\mathbf{W}_{\text{up}} + \hat{\sum}_{\Delta\mathbf{a} \in \mathcal{S}} \Delta\mathbf{a} + \mathbf{b}_{\text{up}}). \tag{8}$$

## 5 Experiment

### 5.1 Evaluation Benchmarks

**Dataset.** We utilize the PEAK-CF and PEAK-T datasets [33], which contain a large number of questions with multiple answers. We keep the questions for which Llama3-8B [32] and Mistral-7B

---

**Algorithm 2** The Edit Composing Stage

---
1: **Input** The initial LLM $f_\theta$, the vector database $\mathcal{S}_{\text{db}}$, the test dataset $\mathcal{D}_{\text{test}}$.
2: **Output** The answers of the queries in $\mathcal{D}_{\text{test}}$.
3: **for** each query $\mathcal{X}_t \in \mathcal{D}_{\text{test}}$ **do**
4:     Retrieve edits and calculate the $\Delta\mathbf{a}$ set and $\mathcal{S}_t$ related to $\mathcal{X}_t$ with its representation $h_{\text{db}}$ (Eq. (9)).
5:     Generate the output with replacing $h_{\text{out}} = \text{FFN}(h_{\text{in}})$ (Eq. (3)) with $h_{\text{out}} = \text{FFN}_{\text{test}}(h_{\text{in}}, \mathcal{S}_t)$ (Eq. (8))
6: **end for**

---

[35] models have more than four unknown answers, with 1,949 instances in PEAK-CF and 922 instances in PEAK-T. We directly use the rephrase instances and locality instances from PEAK-CF and PEAK-T to test the generalization and locality. Please refer to Appendix G for the specific construction process.

**Evaluation metrics.** Inspired by [33], we evaluate baselines and our methods with $SR\text{-}S = \frac{1}{|\mathcal{D}_{\text{test}}|}\sum_{t=1}^{|\mathcal{D}_{\text{test}}|} \mathbb{I}(\forall a \in \mathcal{Y}_t, \text{rank}(a) < |\mathcal{Y}_t|)$, which represents the success rate with all edited answers in the output without *incorrect preceding*. In Appendix H, we provide examples about $SR\text{-}S$ while discussing its effectiveness for evaluation and the results of its variants; $SR\text{-}1 = \frac{1}{c|\mathcal{D}_{\text{test}}|}\sum_{i=1}^{|\mathcal{D}_{\text{test}}|}\sum_{a \in \mathcal{Y}_t} \mathbb{I}(a \in \hat{\mathcal{Y}}_t)$, which represents the success rate that edited answers $a \in \mathcal{Y}_t$ are in the output answer set $\hat{\mathcal{Y}}_t$. $c$ represents the composition number, which means $c$ pieces of edited knowledge are needed for a question; $GSR = \frac{1}{|\mathcal{D}_{\text{rep}}|}\sum_{t=1}^{|\mathcal{D}_{\text{rep}}|} \mathbb{I}(\forall a \in \mathcal{Y}_t, \text{rank}(a) < |\mathcal{Y}_t|)$, but the questions are changed into rephrased ones with the same semantic to evaluate the generalization; $LSR = \frac{1}{|\mathcal{D}_{\text{loc}}|}\sum_{t=1}^{|\mathcal{D}_{\text{loc}}|} \text{Rouge-L}(\hat{\mathcal{Y}}_t^{\text{loc}}, \mathcal{Y}_t^{\text{loc}})$, which represents the f1 score of Rouge-L [36] between the edited outputs $\hat{\mathcal{Y}}_t^{\text{loc}}$ for locality questions and origin output $\mathcal{Y}_t^{\text{loc}}$. Please refer to Appendix K for examples of the input.

**Baselines.** We compared our method with FT, ROME, KE, MEND, AlphaEdit, WISE, T-patcher, GRACE, and MELO. Please refer to Appendix G for more details.

## 5.2 Experimental Results

**Different datasets and CME tasks**. To evaluate the effectiveness of the proposed A³E, we conduct large-scale experiments with all instances of the PEAK-CF and PEAK-T datasets, covering four CME tasks: independent multi-answer composition (IMAC), independent multi-question composition (IMQC), multi-answer composition (MAC) and multi-question composition (MQC). In independent scenarios, we manually filter out useless edits for each question to directly verify the effectiveness of adaptive combination, regularization and merging without the effect of vector database. In MAC and MQC, the vector database is used to evaluate in real-world scenarios with a large number of edits asynchronously and distributedly. We set the composition number to 2, which means two pieces of edited knowledge are needed for a question. The results shown in Table 2 indicate that A³E consistently performs better on four CME tasks across two datasets. For IMAC on PEAK-CF, A³E surpasses all baselines by at least 22.45% in *SR-S*, at least 13.63% in *SR-1* and at least 18.36% in *GSR*. Even on challenging MAC, A³E consistently achieves the best editing performance and generalization while maintaining similar level locality. The lead of A³E on PEAK-T is even more obvious, with at least 67.16% and 406.73% in *SR-S* for IMQC and MQC respectively.

**Different number of edits.** To evaluate the performance of A³E throughout the lifelong usage process, we compare the performance among A³E, GRACE and MELO, which have relatively strong MAC and MQC capability. The results of MAC on PEAK-CF, as shown in Fig. 8(a), indicate that A³E maintains the most outstanding performance until 3898 edits. Please refer to Appendix. J.1 for more results.

**Different composition number.** So far, we have conducted experiments with a composition number of 2. To demonstrate that A³E still has superiority when it needs to simultaneously utilize more edited knowledge, we compared the performance of A³E, GRACE, and MELO with composition numbers ranging from 1 to 4, each containing 50 ($50\times1$), 100 ($50\times2$), 150 ($50\times3$) and 200 ($50\times4$)

Table 2: Performance measured by metrics in Sec. 5.1 in 1) IMAC (3898 edits) and MAC (3898 edits) on PEAK-CF. 2) IMAC (1984 edits) and MAC (1984 edits) on PEAK-T. 3) IMQC (1948 edits) and MQC (1948 edits) on PEAK-CF. 4) IMQC (992 edits) and MQC (992 edits) on PEAK-T.

| Method | PEAK-CF | | | | | | | PEAK-T | | | | | | |
| --- | --- | --- | --- | --- | --- | --- | --- | --- | --- | --- | --- | --- | --- | --- |
| | IMAC | | | MAC | | | | IMAC | | | MAC | | | |
| | SR-S↑ | SR-1↑ | GSR↑ | SR-S↑ | SR-1↑ | GSR↑ | LSR↑ | SR-S↑ | SR-1↑ | GSR↑ | SR-S↑ | SR-1↑ | GSR↑ | LSR↑ |
| FT | 0.31 | 12.01 | 0.36 | 0.00 | 0.00 | 0.00 | 5.25 | 0.10 | 1.92 | 0.10 | 0.00 | 0.00 | 0.00 | 3.79 |
| ROME | 0.92 | 16.30 | 0.82 | 0.00 | 0.62 | 0.00 | 12.25 | 0.30 | 18.24 | 0.60 | 0.00 | 0.00 | 0.00 | 6.22 |
| KE | 0.92 | 10.31 | 0.72 | 0.00 | 0.00 | 0.00 | 10.51 | 0.10 | 5.04 | 0.10 | 0.00 | 0.00 | 0.00 | 4.56 |
| MEND | 2.36 | 23.09 | 1.90 | 0.00 | 0.56 | 0.00 | 12.35 | 1.01 | 12.40 | 0.81 | 0.00 | 0.00 | 0.00 | 5.41 |
| AlphaEdit | 5.44 | 31.01 | 4.57 | 6.98 | 33.24 | 6.16 | 15.17 | 2.02 | 21.78 | 2.12 | 0.50 | 16.28 | 0.41 | 8.45 |
| WISE | 39.99 | 65.20 | 38.60 | 0.10 | 5.65 | 0.10 | 15.21 | 23.08 | 54.64 | 21.37 | 0.20 | 2.92 | 0.20 | 8.40 |
| T-patcher | 26.64 | 52.54 | 12.83 | 0.00 | 0.59 | 0.00 | 1.01 | 20.77 | 43.6 | 18.15 | 0.00 | 0.00 | 0.00 | 0.67 |
| GRACE | 15.20 | 57.44 | 14.53 | 12.99 | 50.16 | 9.75 | 27.19 | 6.55 | 41.93 | 6.05 | 5.65 | 33.07 | 5.04 | 37.54 |
| MELO | 19.46 | 44.33 | 18.47 | 18.22 | 38.51 | 16.68 | 24.13 | 9.48 | 28.94 | 9.07 | 6.96 | 22.94 | 6.56 | 35.98 |
| $A^3E$ | 48.97 | 75.49 | 45.69 | 43.38 | 70.72 | 39.48 | 25.76 | 41.63 | 51.62 | 36.90 | 38.00 | 49.60 | 35.89 | 40.97 |

| Method | PEAK-CF | | | | | | | PEAK-T | | | | | | |
| --- | --- | --- | --- | --- | --- | --- | --- | --- | --- | --- | --- | --- | --- | --- |
| | IMQC | | | MQC | | | | IMQC | | | MQC | | | |
| | SR-S↑ | SR-1↑ | GSR↑ | SR-S↑ | SR-1↑ | GSR↑ | LSR↑ | SR-S↑ | SR-1↑ | GSR↑ | SR-S↑ | SR-1↑ | GSR↑ | LSR↑ |
| FT | 5.64 | 21.65 | 4.82 | 0.00 | 0.00 | 0.00 | 2.09 | 0.00 | 3.23 | 0.40 | 0.00 | 0.00 | 0.00 | 2.12 |
| ROME | 23.49 | 47.08 | 19.38 | 0.00 | 0.41 | 0.00 | 4.52 | 19.15 | 21.92 | 14.52 | 0.00 | 0.00 | 0.00 | 3.63 |
| KE | 1.03 | 15.40 | 0.77 | 0.00 | 0.00 | 0.00 | 3.92 | 4.23 | 8.06 | 3.13 | 0.00 | 0.00 | 0.00 | 2.98 |
| MEND | 5.23 | 22.38 | 5.05 | 0.00 | 0.00 | 0.00 | 3.77 | 14.11 | 17.54 | 8.77 | 0.00 | 0.00 | 0.00 | 4.02 |
| AlphaEdit | 7.69 | 28.57 | 5.64 | 5.33 | 24.92 | 4.21 | 6.92 | 16.33 | 18.44 | 11.09 | 5.65 | 25.11 | 5.04 | 4.11 |
| WISE | 46.56 | 70.11 | 44.51 | 0.0 | 3.23 | 0.0 | 7.28 | 27.62 | 32.77 | 24.60 | 0.20 | 3.33 | 0.20 | 4.40 |
| T-patcher | 33.23 | 60.41 | 31.30 | 0.00 | 0.00 | 0.00 | 0.45 | 26.81 | 31.68 | 25.20 | 0.00 | 0.00 | 0.00 | 0.32 |
| GRACE | 6.56 | 39.02 | 5.23 | 3.59 | 34.61 | 2.87 | 38.71 | 6.05 | 32.06 | 5.54 | 4.84 | 30.55 | 3.63 | 38.59 |
| MELO | 20.31 | 44.16 | 18.47 | 9.13 | 36.16 | 8.21 | 43.22 | 6.25 | 23.49 | 5.85 | 2.82 | 19.86 | 2.02 | 36.25 |
| $A^3E$ | 52.51 | 73.49 | 52.41 | 26.46 | 53.44 | 20.31 | 45.73 | 46.17 | 68.85 | 38.31 | 28.63 | 53.73 | 21.77 | 41.37 |

edits, respectively. As shown in Fig. 8(b), although the *SR-S* gradually decreases with increasing composition number, $A^3E$ always maintains a significantly better performance. Please refer to Appendix. J.1 for more results.

**Non-compositional ME.** A stronger composability should not come at the expense of worse non-compositional ME performance. Therefore, we compare the average non-compositional ME performance of $A^3E$ and baselines. The results, as shown in Fig. 8(c), demonstrate that our method is on par with baselines in *SR-1*, *GSR* and *LSR*. We provide non-compositional results of other datasets in Appendix J.6.

**Different backbones and prompts.** To investigate the impact of different backbones (Llama3-8B and Mistral-7B) and few-shot prompts (please refer to Appendix K) on the proposed method, we conduct experiments in MAC setting with 3898 edits.

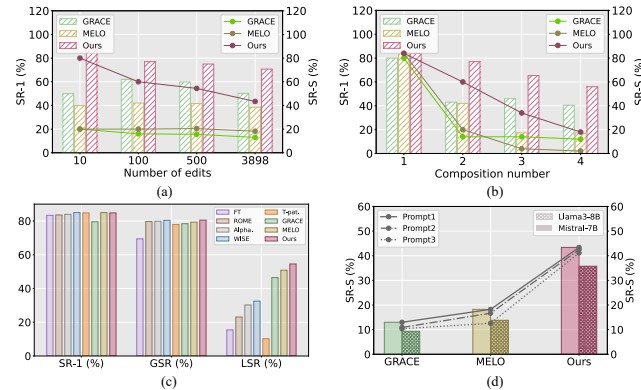

Figure 8: (a) Performance change of $A^3E$ and baselines in MAC until 3898 edits on PEAK-CF. (b) The effect of composition number from 1 to 4 in MAC on PEAK-CF. (c) The comparison between $A^3E$ and baselines in non-compositional ME on PEAK-CF. (d) The effect of different backbones and different prompts in MAC on PEAK-CF.

The results in Fig. 8(d) show that: (1) $A^3E$ has a leading *SR-S* when using different backbones and different prompts. (2) $A^3E$ and baselines generally have better editing performance on Llama3-8B. (3) When using different few-shot prompts to combine multiple questions, there is a certain performance difference in $A^3E$. This may be due to the fact that, although we manually ensure that few-shot prompts do not contain answers and content obviously related to answers, there may still be some implicit associations between them and the answers. However, this does not affect the significant improvement of $A^3E$ on the composability of model editing. Please refer to Appendix. J.3 and Appendix. J.4 for more comprehensive results. We also provide more discussions on failure cases in Appendix J.7 and generalization in Appendix J.8.

## 5.3 Ablation Study

As shown in Fig. 9(a), we conduct experiments in MAC with 3898 edits to evaluate seven ablation versions removing the proposed $\mathcal{L}_o$, $\mathcal{L}_c$, adaptive combination, adaptive merging, norm and editing only $\mathbf{W}_{up}$, respectively. All the components contribute to the composability of $A^3E$. Please refer to Appendix. J.5 for more results. We further demonstrate three advantages of the components.

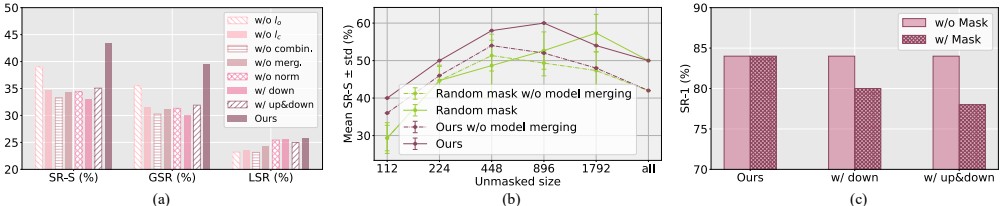

Figure 9: (a) Performance of $A^3E$ and seven ablation versions in MAC on PEAK-CF. (b) The robustness of edited knowledge when randomly masking $\mathbf{B}$ in $\mathbf{W}_{up}$ editing, $\mathbf{W}_{down}$ editing and $\mathbf{W}_{up}\&\mathbf{W}_{down}$ editing in non-compositional ME on PEAK-CF. (c) Performance improved by adaptive merging with random masks from different seeds or adaptive combination in MAC on PEAK-CF.

**Advantage 1.** *Adaptive combination achieves performance beyond the trade-off between composability and learning ability.* As can be seen from Fig. 9(b), although adding a random mask can achieve good composability, the larger standard deviation indicates that it is significantly affected by the random seed. Meanwhile, as the unmask size increases, the learning ability gradually improves with the increasing overlap between edits. We overcome this trade-off by using adaptive combination to select more useful pre-trained foundation knowledge.

**Advantage 2.** $\mathcal{L}_c$ *makes better context-preserving.* See Fig. 5(a), by incorporating $\mathcal{L}_c$, our method significantly reduces the change in the hidden state at the last answer token compared to our methods without $\mathcal{L}_c$, thus improving the context-preserving capability and achieving a level comparable to GRACE. Meanwhile, as shown in Fig. 5(b), our stronger context-preserving capability successfully translates into a higher *SR-S*.

**Advantage 3.** *Adaptive merging and adaptive combination mutually reinforce each other.* In Fig. 9(c), we compare the impact of test-time random masks on $\mathbf{B}$ for editing $\mathbf{W}_{down}$ and $\mathbf{W}_{up}$ in the non-compositional ME scenario, where the mask size is set to the mean size of the parts that require adaptive merging in the CME scenario. It shows that editing $\mathbf{W}_{up}$ is less likely to suffer from forgetting caused by the presence of adaptive merging.

## 6 Conclusion

In this paper, we first define and benchmark the CME task, which aims to evaluate the ability of model editing methods to simultaneously utilize multiple edited knowledge. By revisiting existing model editing methods, we point out their lack of composability manifests as *knowledge loss*, *incorrect preceding* and *knowledge sinking*. To mitigate them, we propose $A^3E$ to train more composable and context-preserving edits and compose edits with enhanced composability. Extensive experiments under various scenarios, backbones, and prompts demonstrate the effectiveness of the proposed $A^3E$.

## Acknowledgements

We thank all the anonymous reviewers for their constructive suggestions on improving this paper. This paper is partially supported by Hong Kong Innovation and Technology Fund (ITF) grant MHP/034/22 and National Natural Science Foundation of China (NSFC) grant 62441236.

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

# Appendices Contents

# A  Notation

Table 3: The definition of the notations.

| Notation | Definition |
|---|---|
| $\mathbf{m}$ | Input & output dimension of feed-forward network (FFN) |
| $\mathbf{n}$ | Hidden dimension of FFN |
| $\mathbf{d}$ | Vocabulary size |
| $\mathbf{W}_{\text{up}}$ | Parameters of the first layer of FFN; $\mathbf{W}_{\text{up}} \in \mathbb{R}^{m \times n}$ |
| $\mathbf{W}_{\text{down}}$ | Parameters of the second layer of FFN; $\mathbf{W}_{\text{down}} \in \mathbb{R}^{n \times m}$ |
| $\mathbf{b}_{\text{up}}$ | Bias vector of the first layer of FFN; $\mathbf{b}_{\text{up}} \in \mathbb{R}^{1 \times n}$ |
| $\mathbf{b}_{\text{down}}$ | Bias vector of the second layer of FFN; $\mathbf{b}_{\text{down}} \in \mathbb{R}^{1 \times m}$ |
| $h_{\text{in}}$ | Input hidden state of FFN; $h_{\text{in}} \in \mathbb{R}^{1 \times m}$ |
| $h_{\text{out}}$ | Output hidden state of FFN; $h_{\text{out}} \in \mathbb{R}^{1 \times m}$ |
| $\mathbf{a}$ | Activation vector after the first layer of FFN; $\mathbf{a} \in \mathbb{R}^{1 \times n}$ |
| $(\mathbf{A}, \mathbf{B})$ | Two learnable low-rank matrices $\mathbf{A} \in \mathbb{R}^{n \times r}(\mathbb{R}^{m \times r})$ and $\mathbf{B} \in \mathbb{R}^{r \times m}(\mathbb{R}^{r \times n})$ to constrain the parameter updates to lie in a low-rank subspace following [1,2] |
| $\mathbf{M}$ | The mask to select pre-trained foundation knowledge in $\mathbf{W}_{\text{down}}$; $\mathbf{M} \in \mathbb{R}^{r \times n}$ |
| $h_{\text{logit}}$ | Output logits of LLM |
| $\mathcal{W}$ | The set of all token IDs |
| $\mathcal{S}$ | The preserved edits for inference by the vector database |
| $a$ | Generated answer of LLM |
| $\hat{\mathcal{Y}}$ | The set of generated answers of LLM |
| $\mathcal{Y}$ | The set of edited answers |

# B  Related Work

**Model Editing.** Model editing has become a popular technique to correct LLMs without the costly re-training. Existing model editing methods are mainly divided into constrained fine-tuning, meta-learning, locating-and-editing, and memory-based methods. Meta-learning methods such as MEND [9] and KE [7] train a hypernetwork to generate model updates, with MALMEN [37] addressing the cancellation effects. Locating-and-editing methods like ROME [8] identify factual associations in the feed-forward layer and update them. Building on this, MEMIT [38] extends it to a batch edit setting, while AlphaEdit [19] projects perturbation onto the null space of the preserved knowledge to achieve better lifelong model editing. To avoid conflicts among numerous edits and the impact on original knowledge, memory-based methods such as SERAC [39], T-patcher [23], GRACE [21], MELO [22] and WISE [20] construct extra working memory in different knowledge preservation forms, from which useless knowledge is filtered out during inference.

**Model Editing Evaluation.** From single edit to batch edit and then to lifelong edit, the evaluation of model editing methods is becoming increasingly close to real-world applications and more challenging in terms of the ability to handle interference between different edits and with the original knowledge. In addition to the classic assessments of reliability [9, 10], generalization [10, 11], and locality [8, 12], Cohen et al. [13] proposed evaluating the ripple effect of model editing, which means a series of knowledge that needs to be changed along with the new knowledge. Ma et al. [14] evaluated the knowledge inserted into the model bidirectionally, while these works [15, 16, 17, 18] assessed the impact of model editing methods on general downstream tasks. Unlike previous works, we are the first to point out that model editing methods should possess composability, namely enabling the model to use multiple edited knowledge simultaneously. On the one hand, our work extends locality to scenarios where multiple edits are needed for the same input (multi-question composition & multi-answer composition), which needs to avoid interference between useful edits. On the other hand, our work expands generalization in two ways. From the input perspective, we explore how to make an edit generalize to scenarios where only part of the input requires the edit (multi-question composition). From the output perspective, we explore how to make an edit generalize to scenarios when there is some content that corresponds to multiple other edits in the output (multi-question composition & multi-answer composition). This work takes model editing a step further towards more complete answers and more complex questions. It is noteworthy that a seemingly related

work [40] considers the success of a single edit in a multi-hop question with reasoning, instead of compositional question/answers that require multiple edits in our focus.

**Model Merging.** Model merging aims to merge the weights of different models while retaining their respective knowledge. However, due to permutation symmetry [41] and disjoint optimization trajectories [42], simply averaging two models can lead to a catastrophic drop in their performance. RegMean [43] proposed a closed-form solution by solving a local linear regression problem for each linear layer, but it requires additional data transmission and inference overhead. Fisher merging [44] weighs the parameters in each model with the Fisher Information Matrix [45, 46], but suffers from memory overhead and conflicts during the merging of multiple models. Recently, since different fine-tuned models initialized from the same pre-trained model effectively share a part of the optimization trajectory, model merging methods based on pre-trained models, such as TIES-Merging [34], task arithmetic in tangent space [47], and ColD Fusion [48], have achieved leading results. Our method draws inspiration from these works but differs in that our proposed compositional model editing aims to enable the model to utilize model merging to use multiple edited knowledge simultaneously during the inference.

## C  Compositional vs. Collect & Edit

**compositional model editing addresses a different scenario** where: during the edit training for "fever" and the edit training for "loss of taste", **they are unaware of each other**. This means we cannot collect all of the knowledge to be edited and then directly compose them at text level. We will explain why this scenario is meaningful below.

**As stated in Sec. 2.1 (cf. Lines 97–99) and illustrated in Fig. 1, we believe that it is necessary to support model editing in a federated manner.** Specifically, when an open-source model is deployed globally, numerous distinct errors may arise simultaneously across different deployed regions. We want these deployers can benefit from the edits of each other.

- In such cases, similar to the assumption that raw data cannot be shared in federated learning due to regulations like GDPR [49], **different deployers are only able to share trained edits**. In such cases, when a deployer receives trained edits from other deployers, it does not know the questions of the edits. **It is impossible to get complete answers during edit training** because:
  1. The deployer has to iterate through all previous edit questions to determine which previous edits need to be composed with the newly received edit to form a complete answer.
  2. However, storing all previous edit questions is **impossible in lifelong model editing**.
- When exchanging raw data is allowed, **local deployers still have to perform immediate model editing to mitigate potential adverse effects without waiting to receive errors discovered by other deployers.** In this scenario, assuming that each of $n$ deployers updates an answer to the same question and exchanges these $n$ different answers with each other.
  - Continuously injecting complete knowledge would require $n$ **times edit training for each deployer at most**.
  - Exchanging independently trained edits would only require 1 **time edit training for each deployer**.

  Training edits independently while ensuring their composability supports model editing in a federated manner **much more efficiently**. This is especially important in the field of model editing, which emphasizes **immediate correction of errors**.

Meanwhile, for **multi-question composition**, compositional model editing is clearly meaningful because,

- Edits for any two different questions may need to be composed during testing.
- Different errors to be edited may be discovered at different time.

Therefore, it is impossible to exhaustively cover all possible composition cases for complete knowledge during the edit training process.

## D    The Design of Vector Database

To enhance the *edit composing* stage, we adapt the vector database in [21, 22] in a simple but effective way. We retrieve useful edited knowledge with a threshold $\gamma$ to ensure locality and a top-$p$ selection to balance the completeness of knowledge and interference from useless edits:

$$
\begin{aligned}
\mathcal{S} = \{(\mathbf{A}, \mathbf{B}) | (\mathbf{K}, \mathbf{A}, \mathbf{B}) \in \mathcal{S}_{\mathrm{db}} \text{ and } h_{\mathrm{db}} \mathbf{K}^\top > \gamma \\
\text{and } h_{\mathrm{db}} \mathbf{K}^\top \in \text{top-}p(h_{\mathrm{db}} \mathbf{K}^\top)\},
\end{aligned}
\tag{9}
$$

where $\mathcal{S}$ represents the retrieved edits from the vector database $\mathcal{S}_{\mathrm{db}}$ with $\mathbf{K} \in \mathbb{R}^{1 \times m}$ as the key and $(\mathbf{A}, \mathbf{B})$ as the value. $\mathbf{K}$ is saved from a selected hidden state of the edit question. $h_{\mathrm{db}} \in \mathbb{R}^{1 \times m}$ represents the selected hidden state of the test question, which are hyperparameters. We use $h_{\mathrm{db}}$ to retrieve edits by matching $\mathbf{K}$. Because of the mask $\mathbf{M}$, we only need to save the unmasked $\mathbf{B}$ and the unmasked position. Please refer to Alg. 1 and Alg. 2 for the construction and utilization of $\mathcal{S}_{\mathrm{db}}$. Please also refer to Alg. 2 for the complete test procedure.

## E    Handling Continuous Editing Situations

When adding an edit for "loss of taste," if we recognize that the existing edit for "fever" is outdated, **our method can easily remove the outdated edit** with the following locate-and-delete algorithm:

> 1. Iterate and mask each edit $(\mathbf{A}, \mathbf{B})$ in the retrieved part $\mathcal{S}$ of the vector database during the generation process $\rightarrow$ locate the edit that leads to the outdated answer.
> 2. Delete the located edit.

We consider a three-answer composition scenario with $3 \times 50$ successful edits to test the effectiveness of the locate-and-delete algorithm above, where each question receives two correct edits and one outdated edit. **The accuracy of deleting the outdated edits is 100%**. We encourage the construction of outdated edits feedback and avoidance mechanisms when deploying $\mathrm{A}^3\mathrm{E}$.

## F    Computational and Memory Overhead

In Tab. 4 and Tab. 5, we provide a more detailed analysis of the computational and memory overhead compared to baselines with Llama3-8B and Llama3-70B from four aspects: **storage overhead**, **trainable parameters**, **extra FLOPs for inference (infer.)**, **extra FLOPs for training (train.)**. We only provide the extra FLOPs for training of T-patcher, GRACE, MELO and $\mathrm{A}^3\mathrm{E}$ because other methods that update the whole matrix or train a hypernetwork have significantly larger training overhead. Specifically,

- **Storage overhead** refers to the additional parameters that need to be stored compared to the pre-trained model for 1000 edits.
- **Trainable parameters** denote the parameters that are updated during training for each edit.
- **Infer.** denotes extra FLOPs compared to the pre-trained model for the inference of $u$ tokens with $v$ input tokens.
- **Train.** denotes extra FLOPs compared to the vanilla forward-backward progress with cross-entropy loss for $w$ training steps of an edit with $z$-token answer.

For **Llama3-8B**,

We also provide the extra FLOPs of the proposed adaptive combination, adaptive regularization and adaptive merging for inference and training:

- Adaptive merging of $\mathrm{A}^3\mathrm{E}$: $1.64u \times 10^{-5}$ GFLOPs
- Adaptive combination of $\mathrm{A}^3\mathrm{E}$: $0.03z$ GFLOPs
- Adaptive regularization of $\mathrm{A}^3\mathrm{E}$: $1.28wz \times 10^{-4} + 1.64w \times 10^{-5}$ GFLOPs

In $\mathrm{A}^3\mathrm{E}$, $w$ is set to 50 and $z < u$ for a instance. The extra FLOPs for training and inference are very small compared to the generation of $u$ tokens with $v$ input tokens with an 8B model, which is about $16u + 16v$ GFLOPs.

Table 4: Computational and Memory Overhead in Llama3-8B.

| Method | Storage Overhead | Trainable Parameters | infer. (GFLOPs) | train. (GFLOPs) |
|---|---|---|---|---|
| FT | 0 | 0.54GB | 0 | - |
| ROME | 1.64GB | 0.54GB | 0 | - |
| KE | 0.72GB | 0.72GB | 0 | - |
| MEND | 1.14GB | 1.14GB | 0 | - |
| AlphaEdit | 1.64GB | 0.54GB | 0 | - |
| WISE | 0.11GB | 0.11GB | $0.27v + 0.27u$ | - |
| T-patcher | 0.07GB | 32.77KB | $0.02u$ | $0.01wz$ |
| GRACE | 0.07GB | 32.77KB | 0.02 | 0 |
| MELO | 0.13GB | 98.30KB | 0.02 | 0 |
| $A^3E$ | 0.07GB | 39.94KB | $0.02 + 4.92u \times 10^{-5}$ | $0.03z + 1.28wz \times 10^{-4} + 1.64w \times 10^{-5}$ |

Table 5: Computational Overhead in Llama3-70B.

| Method | Trainable Parameters | infer. (GFLOPs) | train. (GFLOPs) |
|---|---|---|---|
| FT | 1.88GB | 0 | - |
| ROME | 1.88GB | 0 | - |
| KE | 2.52GB | 0 | - |
| MEND | 2.09GB | 0 | - |
| AlphaEdit | 1.88GB | 0 | - |
| WISE | 0.38GB | $0.94v + 0.94u$ | - |
| T-patcher | 65.54KB | $0.03u$ | $0.02wz$ |
| GRACE | 65.54KB | 0.03 | 0 |
| MELO | 180.22KB | 0.03 | 0 |
| $A^3E$ | 72.71KB | $0.03 + 1.19u \times 10^{-4}$ | $0.09z + 1.28wz \times 10^{-4} + 2.87w \times 10^{-5}$ |

For **Llama3-70B**,

We also provide the extra FLOPs of the proposed adaptive combination, adaptive regularization and adaptive merging for inference and training:

- Adaptive merging of $A^3E$: $1.19u \times 10^{-4}$ GFLOPs
- Adaptive combination of $A^3E$: $0.09z$ GFLOPs
- Adaptive regularization of $A^3E$: $1.28wz \times 10^{-4} + 2.87w \times 10^{-5}$ GFLOPs

In $A^3E$, $w$ is set to 50 and $z < u$ for a instance. The extra FLOPs for training and inference are very small compared to the generation of $u$ tokens with $v$ input tokens with an 70B model, which is about $140u + 140v$ GFLOPs.

# G Experiment Details

For the baselines, we follow the reproduction code and hyperparameters as described in [11]. For $A^3E$, we set the unmasked size $k$ to 896 for PEAK-CF and to 448 for PEAK-T. We set the loss weight $\alpha$ to 8 and $\beta$ to 1 to balance their utilities. We store the output of the 5-th down projection layer at the last subject token as $\mathbf{K}$ and $h_{db}$. We set the edited FFN layer to 31, the learning rate to 0.01, and train each edit for 50 epochs. To better evaluate the priority of different answers within the model, for all baselines and $A^3E$, we assign a penalty of 10 to the answers that have already been generated. How to achieve effective compositional model editing under both instruction-tuned models and conventional inference settings remains an area requiring urgent exploration. For GRACE, MELO and $A^3E$, we use the vector database in Appendix D with the same $p = 4$ and $\gamma = 0.5$. The experiments are conducted on a server with 8 NVIDIA RTX 5880 Ada GPUs.

For the benchmark construction with PEAK-CF as an example: (1) The original PEAK-CF is a model editing dataset containing questions with multiple answers. (2) We queried Llama3-8B and Mistral-7B, selecting questions from PEAK-CF where both models missed four or more answers, along with their corresponding missing answers in PEAK-CF. (3) For multi-answer composition, the retained questions and their randomly selected $c$ missing answers form a test instance, where $c$ is the composition number. (See Tab. 17 for an example) (4) For multi-question composition, we randomly selected $c$ retained questions without repetition and one randomly chosen missing answer per question to form a test instance. (See Tab. 19 for an example)

## H  Discussions of *SR-S*

Fig. 10 shows a correct example and a false example of *SR-S*. *SR-S* aims to accurately measure the interference between different edits to evaluate the composability without considering the case of outputting unrelated answers after all correct answers. This is to mitigate the confounding effect of conventional model editing, where "outputting unrelated answers after correct answers" is also a problem, and warrant fairness in evaluating composability. We report the metric *SR-AS* when "outputting unrelated answers after all correct answers" is considered as an error in Table 6. A$^3$E maintains a clearly superior performance.

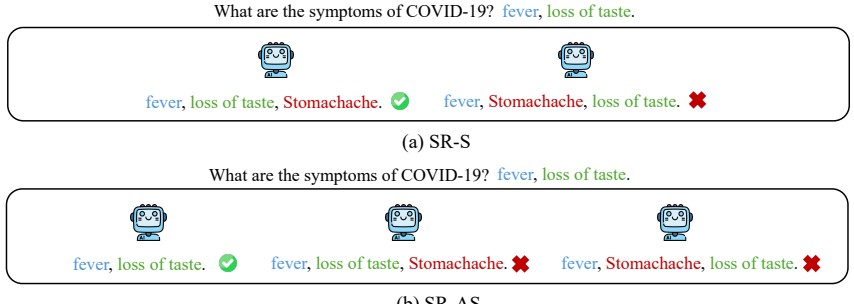

Figure 10: The illustration of (a) *SR-S* and (b) *SR-AS*. The text in blue, red, green represents the first correct answer, the wrong answer and the second correct answer, respectively.

Table 6: Performance measured by *SR-S* and *SR-AS* in 1) IMAC (3898 edits) and MAC (3898 edits) on PEAK-CF. 2) IMAC (1984 edits) and MAC (1984 edits) on PEAK-T. 3) IMQC (1948 edits) and MQC (1948 edits) on PEAK-CF. 4) IMQC (992 edits) and MQC (992 edits) on PEAK-T.

| Method | PEAK-CF | | | | PEAK-T | | | |
| --- | --- | --- | --- | --- | --- | --- | --- | --- |
| | IMAC | | MAC | | IMAC | | MAC | |
| | SR-S↑ | SR-AS↑ | SR-S↑ | SR-AS↑ | SR-S↑ | SR-AS↑ | SR-S↑ | SR-AS↑ |
| FT | 0.31 | 0.26 | 0.00 | 0.00 | 0.10 | 0.00 | 0.00 | 0.00 |
| ROME | 0.92 | 0.62 | 0.00 | 0.00 | 0.30 | 0.05 | 0.00 | 0.00 |
| KE | 0.92 | 0.41 | 0.00 | 0.00 | 0.10 | 0.00 | 0.00 | 0.00 |
| MEND | 2.36 | 1.95 | 0.00 | 0.00 | 1.01 | 0.05 | 0.00 | 0.00 |
| AlphaEdit | 5.44 | 4.67 | 6.98 | 5.23 | 2.02 | 1.64 | 0.50 | 0.30 |
| WISE | 39.99 | 30.12 | 0.10 | 0.00 | 23.08 | 17.97 | 0.20 | 0.10 |
| T-patcher | 26.64 | 18.98 | 0.00 | 0.00 | 20.77 | 16.74 | 0.00 | 0.00 |
| GRACE | 15.20 | 11.90 | 12.99 | 10.31 | 6.55 | 5.13 | 5.65 | 4.31 |
| MELO | 19.46 | 15.03 | 18.22 | 14.73 | 9.48 | 7.60 | 6.96 | 5.54 |
| A$^3$E | **48.97** | **40.69** | **43.38** | **35.92** | **41.63** | **33.68** | **38.00** | **30.90** |

| Method | PEAK-CF | | | | PEAK-T | | | |
| --- | --- | --- | --- | --- | --- | --- | --- | --- |
| | IMQC | | MQC | | IMQC | | MQC | |
| | SR-S↑ | SR-AS↑ | SR-S↑ | SR-AS↑ | SR-S↑ | SR-AS↑ | SR-S↑ | SR-AS↑ |
| FT | 5.64 | 3.70 | 0.00 | 0.00 | 0.00 | 0.00 | 0.00 | 0.00 |
| ROME | 23.49 | 18.99 | 0.00 | 0.00 | 19.15 | 15.12 | 0.00 | 0.00 |
| KE | 1.03 | 0.81 | 0.00 | 0.00 | 4.23 | 3.53 | 0.00 | 0.00 |
| MEND | 5.23 | 3.59 | 0.00 | 0.00 | 14.11 | 10.28 | 0.00 | 0.00 |
| AlphaEdit | 7.69 | 6.16 | 5.33 | 4.31 | 16.33 | 12.70 | 5.65 | 4.54 |
| WISE | 46.56 | 37.68 | 0.00 | 0.00 | 27.62 | 21.67 | 0.20 | 0.00 |
| T-patcher | 33.23 | 26.39 | 0.00 | 0.00 | 26.81 | 20.26 | 0.00 | 0.00 |
| GRACE | 6.56 | 4.93 | 3.59 | 2.77 | 6.05 | 4.54 | 4.84 | 3.93 |
| MELO | 20.31 | 15.61 | 9.13 | 7.39 | 6.25 | 5.04 | 2.82 | 2.28 |
| A$^3$E | **52.51** | **43.02** | **26.46** | **22.07** | **46.17** | **37.40** | **28.63** | **24.19** |

# I The Sufficiency of Editing $W_{up}$

## I.1 For higher *SR-S*

**From an intuitive perspective**, new information does not appear out of thin air, it will be related to past information to some extent.

**From an empirical perspective**, we have shown in Fig. 4(a) that using existing columns to form new knowledge (edit $W_{up}$) can achieve results comparable to or even slightly better than editing $W_{down}$ alone, as well as editing both $W_{up}$ and $W_{down}$ simultaneously. In Fig. 4(b), we conduct the experiment above with fake relation in the input (e.g. share border with -> share boodee with) and fake answer (e.g. Syria -> Sykie) to simulate unseen knowledge. It can be seen that the performance does not change a lot. These experiments indicate that editing $W_{up}$ is enough to guarantee the performance.

**From a mathematical perspective**, $W_{down}$ is often of full rank, which to some extent guarantees its expressive power in its output space.

## I.2 For less *knowledge loss*

**From an intuitive perspective**, the distinction between pretrained knowledge and edited knowledge from existing model editing methods lies in the fact that pretrained knowledge is the combination of pretrained foundation knowledge in $W_{down}$, whereas existing model editing methods directly manipulate the output of $W_{down}$. Meanwhile, we know that pretrained models can leverage pretrained knowledge to answer questions requiring multiple answers, but when using multiple edited knowledge, they exhibit significant knowledge loss. Therefore, we intuitively hypothesize that knowledge loss may partly stem from the fact that edited knowledge is not the combination of pretrained foundation knowledge.

**From an empirical perspective**, we observe that editing $W_{up}$ (i.e., combining pretrained foundation knowledge into edited knowledge) results in less min knowledge loss across 27-31 layers and across baselines compared to editing $W_{down}$ or editing both $W_{up}$ and $W_{down}$ in Tab. 7 while leading to higher *SR-S* and similar non-compositional performance (Fig. 4).

Table 7: Min knowledge loss across 27-31 layers and across baselines when editing $W_{up}$, $W_{down}$, $W_{up}$ and $W_{down}$.

| Method | Min knowledge loss (%) |
|:---:|:---:|
| $W_{up}$ | 60.6 |
| $W_{down}$ | 71.4 |
| $W_{up}$ and $W_{down}$ | 69.7 |

# J Additional Experimental Results

## J.1 The effect of the composition number and the number of edits

In Fig. 11, we provide complete experimental results for the effect of different number of edits and the composition number in MAC and MQC on PEAK-CF and PEAK-T. In PEAK-CF dataset, we also conducted additional experiments by sampling 50 questions with 10+ unknown answers each and evaluated the performance with a composition number of 10.

Table 8: Experimental results on PEAK-CF dataset with composition number 10.

| Method | ROME | AlphaEdit | WISE | T-patcher | GRACE | MELO | $A^3E$ |
|:---:|:---:|:---:|:---:|:---:|:---:|:---:|:---:|
| SR-1 | 2.40 | 4.60 | 9.40 | 6.60 | 8.60 | 12.20 | 17.20 |

As shown in Tab. 8, the experimental results demonstrate that when a large number of edits need to be applied simultaneously, $A^3E$ still significantly outperforms the baselines on the *SR-1* metric, which means $A^3E$ retains more edited knowledge with large composition numbers.

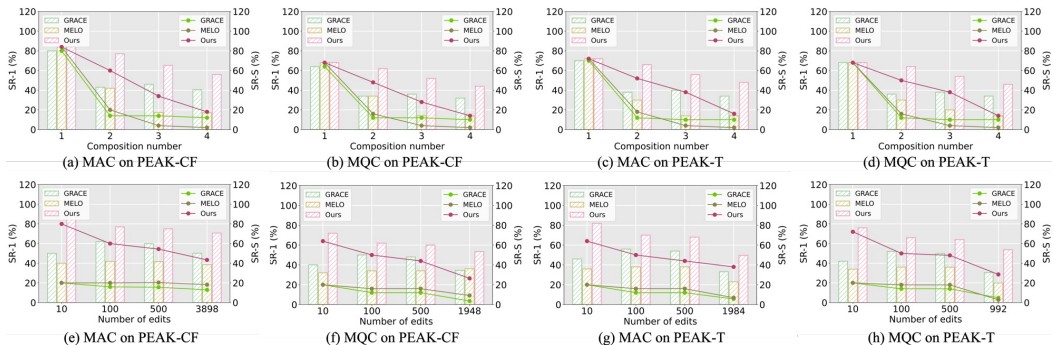

Figure 11: (a)-(d) Performance with different composition number measured by *SR-S* and *SR-1* in MAC and MQC on PEAK-CF and PEAK-T. (e)-(h) Performance change with different number of edits measured by *SR-S* and *SR-1* in MAC and MQC on PEAK-CF and PEAK-T.

## J.2 The effect of hyperparameters

The hyperparameters of $A^3E$ are not sensitive and are easy to adjust. First, as shown in Fig. 12(a)-12(d), compared with the performance improvement of $A^3E$, its performance is not sensitive to the selection of $\alpha$, $\beta$, $\gamma$, and $p$. Second, the adjustment of hyperparameters is straightforward for two reasons: 1) As shown in Fig. 12(a)-12(e), all hyperparameters exhibit similar behavior across different datasets (i.e., PEAK-CF and PEAK-T), indicating that a one-time hyperparameter search is almost sufficient, and repeated re-tuning can be avoided. 2) As shown in Fig. 12(a)-12(b), there exist correlations among certain hyperparameters, e.g., setting $\alpha = 8\beta$ often leads to the best performance. This enables joint tuning, thereby reducing the number of hyperparameters that need to be adjusted.

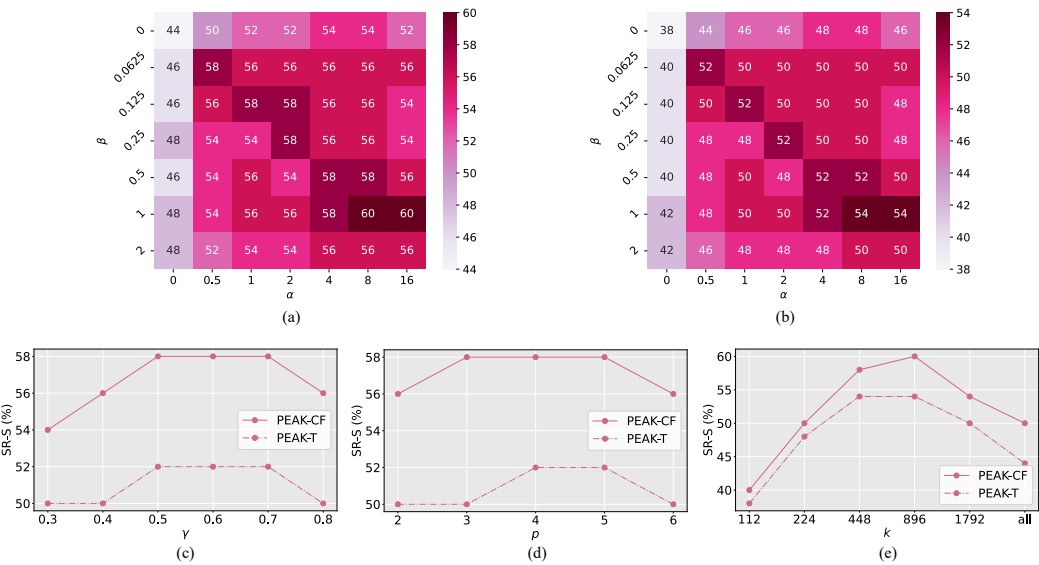

Figure 12: The effect of (a) $\alpha$, $\beta$ on the performance measured by *SR-S* in MAC on PEAK-CF, (b) $\alpha$, $\beta$ on the performance measured by *SR-S* in MAC on PEAK-T. The effect of (c) $\gamma$, (d) $p$ and (e) $k$ on the performance measured by *SR-S* in MAC on PEAK-CF and PEAK-T.

## J.3 The effect of different backbones

Table 9 shows the performance with backbone Mistral-7B in four scenarios on two datasets.

Table 9: Performance with Mistral-7B measured by metrics in Sec. 5.1 in 1) IMAC (3898 edits) and MAC (3898 edits) on PEAK-CF. 2) IMAC (1984 edits) and MAC (1984 edits) on PEAK-T. 3) IMQC (1948 edits) and MQC (1948 edits) on PEAK-CF. 4) IMQC (992 edits) and MQC (992 edits) on PEAK-T.

| METHOD | PEAK-CF | | | | | | | PEAK-T | | | | | | |
| --- | --- | --- | --- | --- | --- | --- | --- | --- | --- | --- | --- | --- | --- | --- |
| | IMAC | | | MAC | | | | IMAC | | | MAC | | | |
| | SR-S↑ | SR-1↑ | GSR↑ | SR-S↑ | SR-1↑ | GSR↑ | LSR↑ | SR-S↑ | SR-1↑ | GSR↑ | SR-S↑ | SR-1↑ | GSR↑ | LSR↑ |
| FT | 0.26 | 10.19 | 0.31 | 0.00 | 0.00 | 0.00 | 4.93 | 0.00 | 2.03 | 0.32 | 0.00 | 0.00 | 0.00 | 2.10 |
| ROME | 0.77 | 13.83 | 0.71 | 0.00 | 0.52 | 0.00 | 11.51 | 14.44 | 17.61 | 11.83 | 0.00 | 0.00 | 0.00 | 3.59 |
| KE | 0.76 | 8.74 | 0.73 | 0.00 | 0.00 | 0.00 | 9.88 | 3.19 | 5.08 | 2.55 | 0.00 | 0.00 | 0.00 | 3.85 |
| MEND | 1.97 | 19.58 | 1.66 | 0.00 | 0.00 | 0.00 | 11.61 | 10.64 | 11.06 | 7.14 | 0.00 | 0.00 | 0.00 | 3.98 |
| ALPHAEDIT | 4.54 | 26.30 | 3.99 | 5.75 | 27.99 | 5.20 | 15.06 | 12.31 | 11.63 | 9.03 | 6.27 | 19.41 | 6.95 | 4.15 |
| WISE | 33.91 | 58.70 | 32.68 | 0.10 | 4.57 | 0.10 | 14.98 | 19.25 | 40.93 | 17.84 | 0.00 | 1.21 | 0.00 | 8.02 |
| T-PATCHER | 22.22 | 44.56 | 11.19 | 0.00 | 0.59 | 0.00 | 0.95 | 20.21 | 31.68 | 20.53 | 0.00 | 0.00 | 0.00 | 0.32 |
| GRACE | 12.68 | 48.72 | 12.68 | 10.70 | 42.24 | 8.22 | 23.55 | 4.56 | 20.22 | 4.51 | 5.37 | 23.61 | 5.00 | 40.14 |
| MELO | 15.24 | 35.25 | 14.52 | 13.80 | 32.99 | 13.23 | 23.11 | 7.86 | 23.99 | 7.60 | 5.75 | 18.95 | 5.24 | 34.76 |
| A³E | 40.84 | 64.03 | 39.86 | 35.73 | 59.55 | 33.30 | 24.21 | 34.80 | 43.43 | 31.21 | 31.75 | 41.53 | 30.04 | 40.88 |

| METHOD | PEAK-CF | | | | | | | PEAK-T | | | | | | |
| --- | --- | --- | --- | --- | --- | --- | --- | --- | --- | --- | --- | --- | --- | --- |
| | IMQC | | | MQC | | | | IMQC | | | MQC | | | |
| | SR-S↑ | SR-1↑ | GSR↑ | SR-S↑ | SR-1↑ | GSR↑ | LSR↑ | SR-S↑ | SR-1↑ | GSR↑ | SR-S↑ | SR-1↑ | GSR↑ | LSR↑ |
| FT | 0.28 | 9.79 | 0.31 | 0.00 | 0.00 | 0.00 | 5.24 | 0.00 | 2.71 | 0.00 | 0.00 | 0.00 | 0.00 | 2.06 |
| ROME | 0.82 | 13.28 | 0.79 | 0.00 | 0.40 | 0.00 | 12.55 | 15.89 | 18.35 | 12.15 | 0.00 | 0.00 | 0.00 | 3.53 |
| KE | 0.82 | 8.40 | 0.69 | 0.00 | 0.00 | 0.00 | 12.49 | 3.51 | 6.75 | 2.62 | 0.00 | 0.00 | 0.00 | 2.90 |
| MEND | 2.12 | 18.82 | 1.82 | 0.00 | 0.36 | 0.00 | 12.73 | 11.71 | 14.69 | 7.34 | 0.00 | 0.00 | 0.00 | 3.92 |
| ALPHAEDIT | 4.88 | 25.26 | 4.38 | 3.60 | 21.03 | 2.82 | 15.70 | 13.55 | 15.44 | 9.28 | 4.74 | 21.01 | 4.44 | 4.00 |
| WISE | 38.96 | 58.73 | 37.27 | 0.00 | 2.71 | 0.00 | 7.32 | 23.39 | 27.42 | 20.56 | 0.00 | 2.82 | 0.00 | 4.20 |
| T-PATCHER | 23.90 | 42.80 | 12.28 | 0.00 | 0.37 | 0.00 | 1.90 | 22.25 | 26.53 | 21.09 | 0.00 | 0.00 | 0.00 | 0.34 |
| GRACE | 13.64 | 46.80 | 13.91 | 6.69 | 31.74 | 4.46 | 44.85 | 5.02 | 26.85 | 4.64 | 4.06 | 25.56 | 3.20 | 37.57 |
| MELO | 16.94 | 36.96 | 15.40 | 7.60 | 30.29 | 6.88 | 42.10 | 5.44 | 19.76 | 4.84 | 2.42 | 16.73 | 1.81 | 36.20 |
| A³E | 43.94 | 61.50 | 43.74 | 22.33 | 44.76 | 18.06 | 45.33 | 38.31 | 57.66 | 32.06 | 23.99 | 44.96 | 19.15 | 40.27 |

## J.4 The effect of different prompts

Table 10-11 shows the performance of A³E and state-of-the-art baselines WISE, GRACE, MELO with different prompts in four scenarios on two datasets.

Table 10: Performance with prompt 2 measured by metrics in Sec. 5.1 in 1) IMAC (3898 edits) and MAC (3898 edits) on PEAK-CF. 2) IMAC (1984 edits) and MAC (1984 edits) on PEAK-T. 3) IMQC (1948 edits) and MQC (1948 edits) on PEAK-CF. 4) IMQC (992 edits) and MQC (992 edits) on PEAK-T.

| Method | PEAK-CF | | | | | | | PEAK-T | | | | | | |
|---|---|---|---|---|---|---|---|---|---|---|---|---|---|---|
| | IMAC | | | MAC | | | | IMAC | | | MAC | | | |
| | SR-S↑ | SR-1↑ | GSR↑ | SR-S↑ | SR-1↑ | GSR↑ | LSR↑ | SR-S↑ | SR-1↑ | GSR↑ | SR-S↑ | SR-1↑ | GSR↑ | LSR↑ |
| FT | 0.26 | 9.97 | 0.30 | 0.00 | 0.00 | 0.00 | 5.36 | 0.09 | 1.59 | 0.09 | 0.00 | 0.00 | 0.00 | 3.15 |
| ROME | 0.76 | 13.54 | 0.68 | 0.00 | 0.51 | 0.00 | 12.17 | 0.50 | 15.15 | 0.25 | 0.00 | 0.00 | 0.00 | 6.17 |
| KE | 0.75 | 8.48 | 0.59 | 0.00 | 0.00 | 0.00 | 10.65 | 0.09 | 4.18 | 0.09 | 0.00 | 0.00 | 0.00 | 4.79 |
| MEND | 1.92 | 18.80 | 1.55 | 0.00 | 0.45 | 0.00 | 12.06 | 0.82 | 10.00 | 0.66 | 0.00 | 0.00 | 0.00 | 5.37 |
| AlphaEdit | 4.60 | 26.21 | 3.86 | 5.90 | 28.10 | 5.21 | 15.82 | 1.71 | 18.41 | 1.80 | 0.42 | 13.76 | 0.34 | 8.14 |
| WISE | 32.81 | 53.54 | 31.67 | 0.13 | 4.59 | 0.12 | 15.46 | 18.88 | 44.78 | 17.72 | 0.27 | 2.51 | 0.27 | 8.04 |
| T-patcher | 22.43 | 44.30 | 10.77 | 0.05 | 0.44 | 0.04 | 1.81 | 17.46 | 36.71 | 15.49 | 0.00 | 0.00 | 0.00 | 0.70 |
| GRACE | 12.66 | 48.04 | 12.10 | 10.88 | 41.96 | 8.21 | 22.73 | 5.39 | 34.98 | 5.22 | 4.62 | 27.81 | 4.32 | 37.57 |
| MELO | 17.75 | 40.52 | 16.84 | 16.68 | 35.20 | 15.31 | 25.05 | 8.58 | 26.35 | 8.45 | 6.26 | 21.11 | 6.11 | 35.07 |
| $A^3E$ | **47.88** | **73.86** | **44.68** | **42.48** | **69.20** | **38.70** | **25.88** | **40.66** | **50.40** | **36.25** | **37.13** | **48.68** | **35.26** | **40.03** |

| Method | PEAK-CF | | | | | | | PEAK-T | | | | | | |
|---|---|---|---|---|---|---|---|---|---|---|---|---|---|---|
| | IMQC | | | MQC | | | | IMQC | | | MQC | | | |
| | SR-S↑ | SR-1↑ | GSR↑ | SR-S↑ | SR-1↑ | GSR↑ | LSR↑ | SR-S↑ | SR-1↑ | GSR↑ | SR-S↑ | SR-1↑ | GSR↑ | LSR↑ |
| FT | 4.68 | 17.98 | 4.00 | 0.00 | 0.00 | 0.00 | 2.73 | 0.01 | 2.68 | 0.34 | 0.00 | 0.00 | 0.00 | 2.76 |
| ROME | 19.51 | 39.10 | 16.09 | 0.00 | 0.34 | 0.00 | 4.75 | 15.92 | 18.20 | 12.07 | 0.00 | 0.00 | 0.00 | 3.02 |
| KE | 0.85 | 12.79 | 0.64 | 0.00 | 0.00 | 0.00 | 3.25 | 3.49 | 6.63 | 2.58 | 0.00 | 0.00 | 0.00 | 2.45 |
| MEND | 4.21 | 18.05 | 4.07 | 0.00 | 0.00 | 0.00 | 3.04 | 11.5 | 14.28 | 7.15 | 0.00 | 0.00 | 0.00 | 4.28 |
| AlphaEdit | 6.50 | 24.15 | 4.76 | 4.51 | 21.06 | 3.56 | 6.85 | 13.81 | 15.58 | 9.38 | 4.77 | 21.23 | 4.26 | 4.48 |
| WISE | 38.14 | 57.51 | 36.46 | 0.09 | 2.55 | 0.08 | 7.90 | 22.51 | 26.66 | 20.52 | 0.06 | 2.96 | 0.06 | 4.88 |
| T-patcher | 27.95 | 50.93 | 26.32 | 0.09 | 0.11 | 0.08 | 0.29 | 22.45 | 26.47 | 21.58 | 0.00 | 0.22 | 0.00 | 0.53 |
| GRACE | 5.35 | 32.55 | 4.25 | 3.10 | 28.88 | 2.48 | 38.33 | 4.87 | 26.57 | 4.94 | 3.83 | 25.81 | 3.24 | 38.58 |
| MELO | 18.45 | 40.30 | 16.78 | 8.45 | 32.99 | 7.60 | 39.48 | 5.65 | 21.22 | 5.53 | 2.36 | 18.40 | 2.05 | 36.45 |
| $A^3E$ | **51.28** | **71.83** | **51.19** | **26.01** | **52.22** | **19.97** | **44.69** | **45.02** | **67.14** | **37.81** | **27.82** | **52.84** | **21.52** | **40.77** |

Table 11: Performance with prompt 3 measured by metrics in Sec. 5.1 in 1) IMAC (3898 edits) and MAC (3898 edits) on PEAK-CF. 2) IMAC (1984 edits) and MAC (1984 edits) on PEAK-T. 3) IMQC (1948 edits) and MQC (1948 edits) on PEAK-CF. 4) IMQC (992 edits) and MQC (992 edits) on PEAK-T.

| Method | PEAK-CF | | | | | | | PEAK-T | | | | | | |
|---|---|---|---|---|---|---|---|---|---|---|---|---|---|---|
| | IMAC | | | MAC | | | | IMAC | | | MAC | | | |
| | SR-S↑ | SR-1↑ | GSR↑ | SR-S↑ | SR-1↑ | GSR↑ | LSR↑ | SR-S↑ | SR-1↑ | GSR↑ | SR-S↑ | SR-1↑ | GSR↑ | LSR↑ |
| FT | 0.28 | 9.33 | 0.24 | 0.00 | 0.00 | 0.00 | 5.08 | 0.08 | 1.49 | 0.08 | 0.00 | 0.00 | 0.00 | 3.95 |
| ROME | 0.72 | 12.78 | 0.64 | 0.00 | 0.48 | 0.00 | 12.61 | 0.47 | 14.31 | 0.24 | 0.00 | 0.00 | 0.00 | 6.88 |
| KE | 0.72 | 8.09 | 0.56 | 0.00 | 0.00 | 0.00 | 10.24 | 0.10 | 3.95 | 0.08 | 0.00 | 0.00 | 0.00 | 4.58 |
| MEND | 1.83 | 17.92 | 1.47 | 0.00 | 0.43 | 0.00 | 12.58 | 0.80 | 9.81 | 0.65 | 0.00 | 0.00 | 0.00 | 5.28 |
| AlphaEdit | 4.34 | 24.78 | 3.65 | 5.58 | 26.56 | 4.92 | 15.12 | 1.62 | 17.40 | 1.62 | 0.40 | 13.01 | 0.33 | 8.75 |
| WISE | 31.58 | 51.53 | 30.48 | 0.13 | 4.42 | 0.12 | 15.99 | 18.17 | 43.10 | 17.06 | 0.26 | 2.42 | 0.26 | 8.78 |
| T-patcher | 20.97 | 41.43 | 10.07 | 0.05 | 0.41 | 0.04 | 1.75 | 16.34 | 34.36 | 14.51 | 0.00 | 0.00 | 0.00 | 0.66 |
| GRACE | 12.05 | 45.75 | 11.52 | 10.41 | 39.95 | 7.82 | 27.64 | 5.13 | 33.30 | 4.97 | 4.40 | 26.49 | 4.12 | 37.07 |
| MELO | 13.36 | 30.52 | 12.68 | 12.57 | 26.51 | 11.55 | 24.03 | 6.44 | 19.82 | 6.41 | 4.69 | 15.94 | 4.63 | 34.95 |
| $A^3E$ | **46.38** | **71.54** | **43.28** | **41.15** | **67.03** | **37.49** | **25.39** | **39.25** | **48.64** | **35.03** | **35.80** | **46.99** | **34.02** | **40.86** |

| Method | PEAK-CF | | | | | | | PEAK-T | | | | | | |
|---|---|---|---|---|---|---|---|---|---|---|---|---|---|---|
| | IMQC | | | MQC | | | | IMQC | | | MQC | | | |
| | SR-S↑ | SR-1↑ | GSR↑ | SR-S↑ | SR-1↑ | GSR↑ | LSR↑ | SR-S↑ | SR-1↑ | GSR↑ | SR-S↑ | SR-1↑ | GSR↑ | LSR↑ |
| FT | 4.38 | 16.81 | 3.74 | 0.00 | 0.00 | 0.00 | 2.62 | 0.01 | 2.50 | 0.32 | 0.00 | 0.00 | 0.00 | 2.65 |
| ROME | 18.42 | 36.93 | 15.20 | 0.00 | 0.32 | 0.00 | 4.54 | 15.03 | 17.19 | 11.40 | 0.00 | 0.00 | 0.00 | 3.85 |
| KE | 0.80 | 12.08 | 0.60 | 0.00 | 0.00 | 0.00 | 3.07 | 3.33 | 6.32 | 2.46 | 0.00 | 0.00 | 0.00 | 2.34 |
| MEND | 4.13 | 17.71 | 3.99 | 0.00 | 0.00 | 0.00 | 3.98 | 10.96 | 13.61 | 6.81 | 0.00 | 0.00 | 0.00 | 4.12 |
| AlphaEdit | 6.14 | 22.83 | 4.50 | 4.26 | 19.91 | 3.36 | 6.53 | 13.06 | 14.73 | 8.87 | 4.51 | 20.07 | 4.02 | 4.29 |
| WISE | 36.71 | 55.35 | 35.09 | 0.09 | 2.45 | 0.08 | 7.67 | 21.66 | 25.65 | 19.77 | 0.36 | 2.86 | 0.20 | 4.74 |
| T-patcher | 26.16 | 47.68 | 24.64 | 0.00 | 0.00 | 0.00 | 0.27 | 20.98 | 24.74 | 20.20 | 0.00 | 0.00 | 0.00 | 0.51 |
| GRACE | 5.09 | 30.99 | 4.04 | 2.96 | 27.49 | 2.37 | 38.78 | 4.63 | 25.28 | 4.72 | 3.64 | 24.59 | 3.10 | 38.04 |
| MELO | 13.87 | 30.34 | 12.61 | 6.39 | 24.84 | 5.74 | 39.73 | 4.26 | 16.13 | 4.11 | 1.89 | 13.76 | 1.45 | 37.08 |
| $A^3E$ | **49.49** | **69.33** | **49.40** | **25.10** | **50.40** | **19.28** | **45.13** | **43.60** | **65.03** | **36.64** | **26.94** | **51.19** | **20.85** | **41.51** |

## J.5 Ablation study

In Table 12, we provide the comprehensive results of ablation study in four scenarios on two datasets.

Table 12: Ablation study measured by metrics in Sec. 5.1 in 1) IMAC (3898 edits) and MAC (3898 edits) on PEAK-CF. 2) IMAC (1984 edits) and MAC (1984 edits) on PEAK-T. 3) IMQC (1948 edits) and MQC (1948 edits) on PEAK-CF. 4) IMQC (992 edits) and MQC (992 edits) on PEAK-T.

| Method | PEAK-CF | | | | | | | PEAK-T | | | | | | |
| | IMAC | | | MAC | | | | IMAC | | | MAC | | | |
| | SR-S↑ | SR-1↑ | GSR↑ | SR-S↑ | SR-1↑ | GSR↑ | LSR↑ | SR-S↑ | SR-1↑ | GSR↑ | SR-S↑ | SR-1↑ | GSR↑ | LSR↑ |
|---|---|---|---|---|---|---|---|---|---|---|---|---|---|---|
| $A^3E$ w/o $l_o$ | 44.07 | 67.94 | 41.12 | 39.04 | 63.65 | 35.53 | 23.19 | 37.47 | 46.46 | 33.21 | 34.20 | 44.64 | 32.30 | 40.97 |
| $A^3E$ w/o $l_c$ | 39.18 | 60.39 | 36.55 | 34.70 | 56.58 | 31.59 | 23.61 | 33.31 | 41.30 | 29.52 | 30.40 | 39.68 | 28.71 | 40.77 |
| $A^3E$ w/o combin. | 37.54 | 57.88 | 35.03 | 33.26 | 54.22 | 30.27 | 23.15 | 32.13 | 39.83 | 28.47 | 29.32 | 38.28 | 27.70 | 40.61 |
| $A^3E$ w/o merg. | 38.60 | 59.51 | 36.02 | 34.20 | 55.75 | 31.12 | 24.31 | 32.47 | 40.26 | 28.78 | 29.64 | 38.69 | 28.00 | 40.95 |
| $A^3E$ w/o norm | 38.85 | 59.89 | 36.25 | 34.41 | 56.10 | 31.32 | 25.44 | 31.85 | 39.49 | 28.23 | 29.07 | 37.95 | 27.46 | 41.34 |
| $A^3E$ w/ down | 37.30 | 57.50 | 34.80 | 33.04 | 53.86 | 30.07 | 25.62 | 31.22 | 38.71 | 27.67 | 28.50 | 37.20 | 26.92 | 40.72 |
| $A^3E$ w/ up&down | 39.58 | 61.02 | 36.93 | 35.07 | 57.16 | 31.91 | 25.76 | 32.54 | 40.35 | 28.84 | 29.70 | 38.77 | 28.06 | 40.02 |
| $A^3E$ | **48.97** | **75.49** | **45.69** | **43.38** | **70.72** | **39.48** | **25.76** | **41.63** | **51.62** | **36.90** | **38.00** | **49.60** | **35.89** | **40.97** |

| Method | PEAK-CF | | | | | | | PEAK-T | | | | | | |
| | IMQC | | | MQC | | | | IMQC | | | MQC | | | |
| | SR-S↑ | SR-1↑ | GSR↑ | SR-S↑ | SR-1↑ | GSR↑ | LSR↑ | SR-S↑ | SR-1↑ | GSR↑ | SR-S↑ | SR-1↑ | GSR↑ | LSR↑ |
|---|---|---|---|---|---|---|---|---|---|---|---|---|---|---|
| $A^3E$ w/o $l_o$ | 47.26 | 66.14 | 47.17 | 23.81 | 48.09 | 18.28 | 45.66 | 41.56 | 61.96 | 34.47 | 25.76 | 48.35 | 19.59 | 39.23 |
| $A^3E$ w/o $l_c$ | 42.01 | 58.79 | 41.93 | 21.17 | 42.75 | 16.24 | 45.59 | 36.94 | 55.07 | 35.65 | 22.90 | 42.98 | 17.42 | 39.10 |
| $A^3E$ w/o combin. | 40.52 | 56.71 | 40.45 | 20.42 | 41.23 | 15.67 | 44.29 | 35.39 | 52.78 | 29.38 | 21.95 | 41.19 | 16.68 | 39.17 |
| $A^3E$ w/o merg. | 40.96 | 57.32 | 40.88 | 20.64 | 41.68 | 15.84 | 44.67 | 36.40 | 54.28 | 30.20 | 22.57 | 42.36 | 17.17 | 39.61 |
| $A^3E$ w/o norm | 40.17 | 56.22 | 40.10 | 20.24 | 40.88 | 15.53 | 44.99 | 36.63 | 54.62 | 30.39 | 22.71 | 42.63 | 17.28 | 40.82 |
| $A^3E$ w/ down | 39.38 | 55.12 | 39.31 | 19.84 | 40.08 | 15.23 | 44.30 | 35.17 | 52.44 | 29.18 | 21.80 | 40.93 | 16.59 | 38.51 |
| $A^3E$ w/ up&down | 41.04 | 57.45 | 40.97 | 20.68 | 41.77 | 15.87 | 44.75 | 37.32 | 55.65 | 30.97 | 23.14 | 43.44 | 17.60 | 38.33 |
| $A^3E$ | **52.51** | **73.49** | **52.41** | **26.46** | **53.44** | **20.31** | **45.73** | **46.17** | **68.85** | **38.31** | **28.63** | **53.73** | **21.77** | **41.37** |

## J.6 Non-compositional ME evaluation on classic datasets for ME

As shown in Tab. 13, the non-compositional performance of $A^3E$ is on par with baselines.

Table 13: Non-compositional results on classic ME dataset ZsRE [50] and Counterfact [8] following [11] on Llama3-8B.

| Method | ZsRE | | | Counterfact | | |
| | SR-1↑ | GSR↑ | LSR↑ | SR-1↑ | GSR↑ | LSR↑ |
|---|---|---|---|---|---|---|
| FT | 47.95 | 52.32 | 73.08 | 51.53 | 48.14 | 63.95 |
| ROME | 99.53 | 97.52 | 95.57 | 100.00 | 92.77 | 77.40 |
| MEND | 92.24 | 90.70 | 96.23 | 79.10 | 58.76 | 91.19 |
| AlphaEdit | 98.00 | 96.54 | 95.57 | 99.55 | 95.03 | 79.21 |
| WISE | 93.39 | 92.53 | 100.00 | 98.59 | 93.50 | 99.32 |
| T-patcher | 95.28 | 94.24 | 92.62 | 95.03 | 93.79 | 89.38 |
| GRACE | 96.69 | 94.00 | 100.00 | 99.77 | 94.12 | 100.00 |
| MELO | 95.28 | 94.08 | 100.00 | 98.53 | 92.88 | 100.00 |
| $A^3E$ | 97.00 | 96.77 | 100.00 | 99.77 | 93.79 | 100.00 |

## J.7 Failure cases analysis

We conducted **failure case analysis for overlapping, contradictory, and capacity-exceeding edits** as follows:

- For **overlapping and contradictory cases**, we conducted statistical analysis on the PEAK-CF dataset under the IMAC scenario with 50 cases of two-edit composition, measuring:
  - The overlapping selection of pre-trained foundation knowledge, calculated by the overlapping length between non-zero $\Delta a$ (Sec. 4.1) regions of two edits in adaptive merging (Sec. 4.2)
  - The contradictory utilization of pre-trained foundation knowledge, calculated by the length of regions with different signs between $\Delta a$ (Sec. 4.1) of two edits in adaptive merging (Sec. 4.2)

The table below presents how the *SR-S* metric, which is inversely proportional to the proportion of failure cases, varies across different overlapping and contradictory ranges. As the overlapping length and contradictory length increase, the *SR-S* of $A^3E$ gradually decreases, but it remains significantly higher than the baselines.

Table 14: *SR-S* across different overlapping ranges.

| Overlapping | $[0, 500)$ | $[500, 1000)$ | $[1000, 1500)$ | $[1500, +\infty)$ |
|---|---|---|---|---|
| SR-S | 75.00 | 66.67 | 45.45 | 50.00 |

Table 15: *SR-S* across different contradictory ranges.

| Contradictory | $[0, 200)$ | $[200, 400)$ | $[400, 600)$ | $[600, +\infty)$ |
|---|---|---|---|---|
| SR-S | 66.67 | 68.75 | 57.14 | 46.67 |

Table 16: *SR-S* comparison with baselines.

| Baselines | MEND | ROME | AlphaEdit | WISE | T-patcher | GRACE | MELO |
|---|---|---|---|---|---|---|---|
| SR-S | 4.00 | 4.00 | 8.00 | 32.00 | 20.00 | 14.00 | 20.00 |

- For **capacity-exceeding cases**, we found that the failure cases of baselines and $A^3E$ in both compositional model editing and non-compositional model editing scenarios often involve numerical data in answers. For example,

    - **Edit 1**: Rheinmetall, which has designed **12 cm leFH 08**
    - **Test 1**: Rheinmetall, which has designed **12 cm leFH 8** (failure)
    - **Edit 2**: Rheinmetall, which has designed **35.5 cm Haubitze M1**
    - **Test 2**: Rheinmetall, which has designed **35.5 cm Haubitze M1** (success)
    - **Composition test**: Rheinmetall, which has designed **12 cm leFH**, **12 cm Haubitze M8** (failure)

This may stem from the model's inherent inability to accurately distinguish numerical data. Therefore, addressing such inherent limitations of the model is a valuable direction for future work in model editing.

We believe these findings can guide future research to further improve edit composability.

## J.8 Generalization

The potential generalization limitations are as follows:

- **Noisy scenarios**. In Fig. 8(d), we tested the impact of different few-shot prompts on $A^3E$'s performance. These varied few-shot prompts not only controlled the model's output but also simulated different noise present in the context. The results show that while $A^3E$'s performance is affected, it still maintains a leading performance compared to the baselines.
- **Noisy open-domain scenarios**. We humbly acknowledge that in open-domain scenarios, noisy contexts may still impact the retrieval accuracy of the vector database. Since $A^3E$ primarily focuses on edit training and edit composing, we will explore

    - optimizing the hidden state positions for retrieval based on the real data distribution in application scenarios
    - employing more powerful embedding models
    - training a question-rewriting module to better align with the vector database

    for optimal performance in the future work.
- **Adversarial scenarios**. We conducted additional experiments using $2 \times 50$ successful edits from the PEAK-CF dataset under the IMAC setting by providing the tested question with incorrect answers in the few-shot prompts. The results demonstrate that $A^3E$ consistently (100%) outputs the edited answers across all test cases, proving its robustness against contextual interference.

Table 17: An example of multi-answer composition containing two pieces of edited knowledge, where the blue text indicates the question and the rephrased question.

| | |
|---|---|
| Question | Bertrand Russell is the author of |
| Answer | Power: A New Social Analysis, A History of Western Philosophy |
| Edit 1 $x_e$ | Bertrand Russell is the author of |
| Edit 1 $y_e$ | Power: A New Social Analysis |
| Edit 2 $x_e$ | Bertrand Russell is the author of |
| Edit 2 $y_e$ | A History of Western Philosophy |
| Test $\mathcal{X}_t$ | Tim Dorsey, who has written the Cadillac Beach, Nuclear Jellyfish, Triggerfish Twist, Hammerhead Ranch Motel, The Big Bamboo, Orange Crush (novel), Hurricane Punch, The Stingray Shuffle, Atomic Lobster, Torpedo Juice (novel), Florida Roadkill.\n Jerusalem, which is the partner town of NYC, New York, Praha, New York City, United States, Rio de Janeiro, NY, New York, NY, Prague, New York City, Tehran, Buenos Aires, Moscow, Manhattan.\n Pushkin is the author of The Fountain of Bakhchisaray, Eugene Onegin, The Tale of the Fisherman and the Fish, Poltava (poem), The Tale of the Golden Cockerel, Dubrovsky (novel), The Belkin Tales, Onegin, The Stone Guest (play), The Bronze Horseman (poem), The Queen of Spades (story), The Tale of the Priest and of His Workman Balda, The Gypsies, The Blizzard, Tatiana Larina, The Tale of the Dead Princess and the Seven Knights.\n WWE is the owner of WWE Classics on Demand, FCW Florida Heavyweight Championship, WWE Studios, WWE Films, WWE Network, NXT, FCW, WWE Classics On Demand, FCW Southern Heavyweight Championship, WCW, World Championship Wrestling, WCW, Inc., Florida Championship Wrestling, NXT Wrestling, Universal Wrestling Corporation, WWE NXT.\n Bertrand Russell is the author of |
| Test $\mathcal{Y}_t$ | Power: A New Social Analysis, A History of Western Philosophy |
| Rephrased test | Tim Dorsey, who has written the Cadillac Beach, Nuclear Jellyfish, Triggerfish Twist, Hammerhead Ranch Motel, The Big Bamboo, Orange Crush (novel), Hurricane Punch, The Stingray Shuffle, Atomic Lobster, Torpedo Juice (novel), Florida Roadkill.\n Jerusalem, which is the partner town of NYC, New York, Praha, New York City, United States, Rio de Janeiro, NY, New York, NY, Prague, New York City, Tehran, Buenos Aires, Moscow, Manhattan.\n Pushkin is the author of The Fountain of Bakhchisaray, Eugene Onegin, The Tale of the Fisherman and the Fish, Poltava (poem), The Tale of the Golden Cockerel, Dubrovsky (novel), The Belkin Tales, Onegin, The Stone Guest (play), The Bronze Horseman (poem), The Queen of Spades (story), The Tale of the Priest and of His Workman Balda, The Gypsies, The Blizzard, Tatiana Larina, The Tale of the Dead Princess and the Seven Knights.\n WWE is the owner of WWE Classics on Demand, FCW Florida Heavyweight Championship, WWE Studios, WWE Films, WWE Network, NXT, FCW, WWE Classics On Demand, FCW Southern Heavyweight Championship, WCW, World Championship Wrestling, WCW, Inc., Florida Championship Wrestling, NXT Wrestling, Universal Wrestling Corporation, WWE NXT.\n Bertrand Russell is the writer of |
| Locality test | Philosophy, which is played by |

- **Larger models and black-box models**. A$^3$E is applicable to larger LLMs. Due to limitations in computational resources, we provide experimental results for GRACE, MELO, and A$^3$E under the MAC scenario using Llama3-70B and the PEAK-CF dataset. As shown in Table 18, A$^3$E maintains the leading performance.

Table 18: MAC and MQC performance on PEAK-CF with Llama3-70B.

| Method | MAC | | | | MQC | | | |
|---|---|---|---|---|---|---|---|---|
| | SR-S↑ | SR-1↑ | GSR↑ | LSR↑ | SR-S↑ | SR-1↑ | GSR↑ | LSR↑ |
| GRACE | 16.32 | 42.48 | 11.24 | 24.02 | 3.59 | 37.28 | 3.62 | 38.16 |
| MELO | 28.12 | 48.64 | 26.58 | 23.18 | 12.13 | 39.27 | 11.33 | 41.41 |
| A$^3$E | 50.69 | 69.47 | 47.10 | 25.10 | 33.12 | 55.65 | 32.98 | 44.26 |

However, A$^3$E cannot be applied to black-box models. In the field of knowledge editing for large language models, one research direction focuses on editing white-box models, i.e., model editing [22, 21, 20, 19]. The access to model weights is a prerequisite for applying these editing techniques. Another research direction addresses knowledge editing for extremely large-scale models where model editing may be prohibitively expensive or black-box models by performing edits at the text level, including [51, 52, 13, 40, 53, 54, 55, 56, 57, 58, 59, 60]. A$^3$E follows the first

direction to edit white-box models and focuses on compositional model editing because text-level knowledge editing is not applicable for the following reason:

- As stated in Sec. 2.1 and illustrated in Fig. 1, we aims to support model editing in a federated manner [61, 62, 63, 64]. Specifically, when a model is deployed globally, numerous distinct errors may arise simultaneously across different deployed regions. We want these deployers can benefit from the edits of each other.

- In such cases, similar to the assumption that raw data cannot be shared in federated learning due to regulations like GDPR [49], different deployers are only able to share trained edits rather than the texts.

In a centralized scenario, $A^3E$ can also be implemented by the owner of black-box models for immediate error correction with much more composable edits. We recommend users to choose methods from either direction based on their needs, and we will explore how to perform compositional knowledge editing at the text level for black-box models in our future work.

## K  Description of Dataset and Few-shot Prompts

In Table 17 and 19, we provide examples of the simultaneous use of two pieces of knowledge in the MAC and MQC scenarios, respectively. In Table 20, we provide another two prompts to test the effect of different prompts.

Table 19: An example of multi-question composition containing two edited pieces of knowledge, where the blue text indicates the question and the rephrased question.

| Edit 1 $x_e$ | Carmarthenshire shares border with |
|---|---|
| Edit 1 $y_e$ | Ceredigion |
| Edit 2 $x_e$ | Turkey shares border with |
| Edit 2 $y_e$ | Syria |
| Test $\mathcal{X}_t$ | The answers of the questions "Daimler has made the", "Saxony is adjacent to" are Mercedes-Benz, Hamburg respectively.\n The answers of the questions "Karlheinz Stockhausen is the composer of", "Daimler is the parent organization of" are Originale, Car2go respectively.\n The answers of the questions "Katherine Roberts is the writer of", "Florida International University has the employer" are The Colossus Crisis, Carlos Alvarez respectively.\n The answers of the questions "contraception has a subclass of", "Kering has subsidiary" are Intrauterine device, Volcom respectively.\n The answers of the questions "Carmarthenshire shares border with", "Turkey shares border with" are |
| Test $\mathcal{Y}_t$ | Ceredigion, Syria |
| Rephrased test | The answers of the questions "Daimler has made the", "Saxony is adjacent to" are Mercedes-Benz, Hamburg respectively.\n The answers of the questions "Karlheinz Stockhausen is the composer of", "Daimler is the parent organization of" are Originale, Car2go respectively.\n The answers of the questions "Katherine Roberts is the writer of", "Florida International University has the employer" are The Colossus Crisis, Carlos Alvarez respectively.\n The answers of the questions "contraception has a subclass of", "Kering has subsidiary" are Intrauterine device, Volcom respectively.\n The answers of the questions "Carmarthenshire is adjacent to", "Turkey is adjacent to" are |
| Locality test | Vasif Talibov is a citizen of |

## L  Limitations

Although the proposed method significantly enhances the composability of model editing, it still falls short of achieving fully accurate composition. Future research will continue to explore the factors that influence composability and further improve the effectiveness of compositional model editing. The performance of instruct models under penalty-free decoding condition also needs to be explored. Additionally, as this paper primarily focuses on how to train and compose edits with composability, it does not delve deeply into the organization of vector databases. Future work will aim to continuously

Table 20: Different few-shot prompts.

| | |
|---|---|
| Prompt 1 $\mathcal{X}_t$ | Tim Dorsey, who has written the Cadillac Beach, Nuclear Jellyfish, Triggerfish Twist, Hammerhead Ranch Motel, The Big Bamboo, Orange Crush (novel), Hurricane Punch, The Stingray Shuffle, Atomic Lobster, Torpedo Juice (novel), Florida Roadkill.\n Jerusalem, which is the partner town of NYC, New York, Praha, New York City, United States, Rio de Janeiro, NY, New York, NY, Prague, New York City, Tehran, Buenos Aires, Moscow, Manhattan.\n Pushkin is the author of The Fountain of Bakhchisaray, Eugene Onegin, The Tale of the Fisherman and the Fish, Poltava (poem), The Tale of the Golden Cockerel, Dubrovsky (novel), The Belkin Tales, Onegin, The Stone Guest (play), The Bronze Horseman (poem), The Queen of Spades (story), The Tale of the Priest and of His Workman Balda, The Gypsies, The Blizzard, Tatiana Larina, The Tale of the Dead Princess and the Seven Knights.\n WWE is the owner of WWE Classics on Demand, FCW Florida Heavyweight Championship, WWE Studios, WWE Films, WWE Network, NXT, FCW, WWE Classics On Demand, FCW Southern Heavyweight Championship, WCW, World Championship Wrestling, WCW, Inc., Florida Championship Wrestling, NXT Wrestling, Universal Wrestling Corporation, WWE NXT.\n |
| Prompt 2 $\mathcal{X}_t$ | Saxony is adjacent to Hamburg, Nordsachsen, Liberec Region, Sachsen, Saxony-Anhalt, Bavaria, Sachsen-Anhalt, Thuringen, North Rhine-Westphalia, Thuringia, Saxony Anhalt, Lower Saxony, Brandenburg.\n Florida International University has the employer Carlos Alvarez, Les Standiford, Elizabeth Price Foley, Stanley Fish, Leonard Strickman, Barbara Walsh, Jerry Markham.\n Contraception has a subclass of Intrauterine device, withdrawal method, Emergency contraception, Coitus interruptus, Condom, intrauterine device, morning-after pill.\n Kering has subsidiary Volcom, Saint Laurent Paris, Gucci Group, Christopher Kane, Puma, Boucheron, Gucci, Saint Laurent, Bottega Veneta, Balenciaga, Sergio Rossi.\n |
| Prompt 3 $\mathcal{X}_t$ | Westchester County is adjacent to Bronx County, The Bronx, south Bronx, Putnam County, Rockland, Bronx.\n Margery Allingham is the writer of The Case of the Late Pig, Hide My Eyes, The Mind Readers, Cargo of Eagles, The China Governess, Sweet Danger, Coroner's Pidgin.\n The position Philippine President is held by Fidel Ramos, Corazon Aquino, Emilio Aguinaldo, Macapagal, Joseph Ejercito Estrada, Roxas, Quezon, Benigno Aquino III, Diosdado Macapagal, Ferdinand Marcos, Manuel Quezon.\n cytokine has a subclass of Monokine, Interferon, lymphokine, interleukins, Chemokine.\n |

optimize the structure of vector databases. The effectiveness of other PEFT methods [65, 66, 67, 68] need to be explore as well.

## M Broader Impact

This paper presents work whose goal is to advance the field of model editing. Model editing is a cutting-edge concept that holds immense significance in a variety of critical and rapidly evolving fields, including but not limited to medical or legal fields. In these scenarios, mistakes of models are found after deployment, thus a timely and effective correction is needed. Our research has taken a step towards more available model editing by allowing multiple edits to be used simultaneously, leading to safer and more ethical applications across a broad spectrum of industries.

