# OpenReview forum: "A$^3$E: Towards Compositional Model Editing"
_NeurIPS.cc/2025/Conference — NeurIPS 2025 poster_

### Official Review · Reviewer_dHwZ · 2025-06-13

**Clarity:** 2
**Significance:** 2
**Originality:** 3
**Rating:** 4
**Confidence:** 4

**Summary:**

The paper introduces Compositional Model Editing (CME), where multiple edits must be integrated into LLMs. Existing methods fail due to knowledge loss, incorrect precedence, and knowledge sinking. The authors propose A³E, an adaptive editor that combines and merges edits effectively, improving compositional performance by 22.45% without hurting single-edit accuracy.

**Questions:**

I have covered the questions I want to ask in the weakness section. I would increase my rating if the authors can address the concerns.

**Ethical Concerns:**

["NO or VERY MINOR ethics concerns only"]

**Final Justification:**

Thanks to the authors for their detailed responses. My main concern about the motivation of compositional editing has been addressed, so I would like to increase my rating.

**Limitations:**

Yes

**Quality:**

3

**Strengths And Weaknesses:**

Strengths:

- The compositional model editing is a interesting direction and worth to explore.
- They propose a new benchmark for compositional model editing.
- They investigate the issues of existing model editing techniques.

Weakness:

- The first thing that comes to mind is **when** we need composition and **when** we don’t. In some cases, composition is necessary; in others, we may need to revert previously edited knowledge. A better approach for compositional model editing might be to inject complete knowledge into the model and erase prior incomplete knowledge. For example:
  - **Q**: What are the symptoms of COVID-19?
    - **Edit 1**: "Fever"
    - **New correct edit**: "Fever, loss of taste" (rather than adding "loss of taste" as a separate, independent edit)

- How do you handle continuous editing situations, when previously edited knowledge needs to be changed? For example:
  - **Q**: What are the symptoms of COVID-19?
    - **Edit 1**: "Fever"
    - Then, we want to update the answer to "loss of taste" (not "Fever, loss of taste")

- Additionally, I’m not sure how you evaluate the baseline methods, since the details are not included in the paper or appendix. I assume you perform two sequential edits and test whether the model can combine both. If that’s the case, I don’t think it’s a fair comparison, since most baseline methods would treat the latest edit as the new ground truth rather than combining edits. As mentioned in the first point, a better editing strategy would be to erase the previous, incomplete knowledge and then insert the updated, correct information.

---

> ### Author Rebuttal · Authors · 2025-07-31
>
> We sincerely thank Reviewer dHwZ for the valuable comments. Below, please find our responses to each concern, and let us know if any issues remain.
>
> **W1**. The first thing that comes to mind is when we need composition and when we don’t. In some cases, composition is necessary; in others, we may need to revert previously edited knowledge. A better approach for compositional model editing might be to inject complete knowledge into the model and erase prior incomplete knowledge.
> > We greatly appreciate the reviewer for taking this point into consideration, which means that the reviewer has given in-depth thought to the significance of the compositional model editing scenario. In fact, we also took this into account in our research but finally chose to perform each edit independently for the following reasons, which we promise to discuss in the revised version:
> >
> > + The reviewer's example actually falls under standard **non-compositional model editing**, not **compositional model editing**. This is because its assumption is that when editing the question "What are the symptoms of COVID-19?", we already know the standard answer is "fever, loss of taste." The example provided by the reviewer can be resolved by A$^3$E in two steps:
> >   1. Locate and delete the outdated edit "fever" (refer to W2 for details).
> >   2. Update the answer to "fever, loss of taste" with non-compositional model editing.
> >
> >   We demonstrated in Fig. 8\(c\) that the performance of A$^3$E in non-compositional model editing is on par with baselines.
> > + However, **compositional model editing addresses a different scenario** where: during the edit training for "fever" and the edit training for "loss of taste", **they are unaware of each other**. We will explain why this scenario is meaningful below.
> > + **As stated in Sec. 2.1 (cf. Lines 97–99) and illustrated in Fig. 1, we believe that it is necessary to support model editing in a federated manner.** Specifically, when an open-source model is deployed globally, numerous distinct errors may arise simultaneously across different deployed regions. We want these deployers can benefit from the edits of each other.
> >   - In such cases, similar to the assumption that raw data cannot be shared in federated learning due to regulations like GDPR [1], **different deployers are only able to share trained edits**. In such cases, when a deployer receives trained edits from other deployers, it does not know the questions of the edits. **It is impossible to get complete answers during edit training** because:
> >     1. The deployer has to iterate through all previous edit questions to determine which previous edits need to be composed with the newly received edit to form a complete answer.
> >     2. However, storing all previous edit questions is **impossible in lifelong model editing**.
> >   - When exchanging raw data is allowed, **local deployers still have to perform immediate model editing to mitigate potential adverse effects without waiting to receive errors discovered by other deployers.** In this scenario, assuming that each of $n$ deployers updates an answer to the same question and exchanges these $n$ different answers with each other.
> >     - Continuously injecting complete knowledge would require **$n$ times edit training for each deployer at most**.
> >     - Exchanging independently trained edits would only require **$1$ time edit training for each deployer**.
> >
> >     Training edits independently while ensuring their composability supports model editing in a federated manner **much more efficiently**. This is especially important in the field of model editing, which emphasizes **immediate correction of errors**.
> >
> > + Meanwhile, for **multi-question composition**, compositional model editing is clearly meaningful because,
> >   - Edits for any two different questions may need to be composed during testing.
> >   - Different errors to be edited may be discovered at different time.
> >
> >   Therefore,
> >   - It is impossible to exhaustively cover all possible composition cases for complete knowledge during the edit training process.
> >
> > [1] The eu general data protection regulation (gdpr)
>
> **W2**. How do you handle continuous editing situations, when previously edited knowledge needs to be changed?
> > We greatly appreciate the reviewer for drawing our attention to this point, which indeed increases the practicality of A$^3$E. When adding the edit for "loss of taste," if we recognize that the existing edit for "fever" is outdated, **our method can easily remove the outdated edit** with the following locate-and-delete algorithm:
> >
> > 1. Iterate and mask each edit $(\mathbf{A},\mathbf{B})$ in the retrieved part $\mathcal{S}$ of the vector database during the generation process -> locate the edit that leads to the outdated answer.
> > 2. Delete the located edit.
> >
> > We consider a three-answer composition scenario with $3\times50$ successful edits to test the effectiveness of the locate-and-delete algorithm above, where each question receives two correct edits and one outdated edit. **The accuracy of deleting the outdated edits is 100%**. We will encourage the construction of outdated edits feedback and avoidance mechanisms when deploying A$^3$E.
>
> **W3**. The concern about unfair comparison with baselines.
> > We would like to humbly clarify that **our evaluation is fair for evaluating the composability**.
> > + **We adapt all baselines to make it applicable to multi-answer composition without treating the latest edit as the new ground truth.** As shown in Fig. 1, our evaluation of all baselines follows the protocol of **training multiple edits for one question independently and then composing them to ensure a fair comparison, focusing on evaluating the composability**. Please refer to Sec. 2.2 for the edit composing of the baselines. Specifically,
> >   - Each edit is trained without knowledge of other edits for the same question. In IMAC (cf. Lines 269-274), to accurately evaluate the improvement in composability brought by A$^3$E, edits for the same question are trained on top of **the pre-trained model** and then composed. In MAC (cf. Lines 269-274), to verify that we also significantly improve composability in lifelong scenarios, edits for the same question are trained on top of **previous edits for different questions** and then composed. In this way, we adapt the baselines to compose edits. We elaborate on the necessity of this evaluation scenario in W1.
> > + Regarding the reviewer’s suggestion to “erase the previous, incomplete knowledge and then insert the updated, correct information,” we provide a detailed discussion in W1 on why this approach is infeasible for compositional model editing.

---

> > ### Comment · Reviewer_dHwZ · 2025-08-05
> >
> > Thanks to the authors for their detailed responses. My main concern has been addressed, so I would like to increase my rating.

---

> > > ### Author Response · Authors · 2025-08-06
> > >
> > > Dear Reviewer dHwZ,
> > >
> > > Thanks for raising the score. We are truly pleased to see that your concerns have been addressed. Your in-depth thought to the significance of the compositional model editing scenario is greatly appreciated. We will integrate the suggestions and discussions into our revised manuscript.

---

### Official Review · Reviewer_cjhv · 2025-07-01

**Clarity:** 2
**Significance:** 3
**Originality:** 3
**Rating:** 4
**Confidence:** 5

**Summary:**

This paper proposes Compositional Model Editing (CME), where large language models must integrate multiple knowledge edits to answer complex, real-world questions. Existing model editing methods are shown to struggle in this setting due to issues like knowledge loss and incorrect preceding. To address this, the authors introduce $A^3E$, a two-stage editing framework that improves edit composability through adaptive combination, regularization, and merging techniques. Extensive experiments demonstrate that $A^3E$ significantly outperforms prior methods in compositional settings without reducing standard editing performance.

**Questions:**

1. Could you clarify the distinction between Incorrect Preceding and Knowledge Sinking?
2. How is the adaptive mask in the Adaptive Combination module computed in practice?
3. You propose setting conflicting $\delta a$ elements to zero. Could you clarify do you apply this merging at the level of each token, each edit, or batch-wise? I get a little lost in this part.

**Ethical Concerns:**

["NO or VERY MINOR ethics concerns only"]

**Final Justification:**

I appreciate the clarifications and additional experiments and hope the authors can incorporate these helpful clarifications and improvements in the final version of the paper. I'll keep my scores.

**Limitations:**

No.

Suggestions for Improvement:
1. Discuss generalization limits. It would be helpful to mention how it might perform in more noisy, open-domain, or adversarial scenarios.
2. The paper lacks a thorough analysis of how the method scales to larger numbers of edits per query (e.g., 10 or more).
3. The performance is much higher than the baselines. I suggest you check if there are any mistakes.

**Quality:**

3

**Strengths And Weaknesses:**

Strengths
1. The paper identifies gaps between current model editing and real-world needs by proposing the Compositional Model Editing (CME) task, which is both novel and practically important.
2. $A^3E$'s architecture thoughtfully aligns with the challenges identified. Each of the three components (adaptive combination, regularization, and merging) corresponds to the failures of prior works.
3. The method shows large and consistent improvements in compositional settings while preserving performance in non-compositional tasks.

Weaknesses:
1. While Figure 7 summarizes the core components of $A^3E$, it presents them in a very abstract and schematic way. The figure lacks concrete examples or intuitive annotations, which makes it difficult to understand how each module actually works in practice. Adding a running example would make the method significantly more accessible.
2. In the introduction, terms like Incorrect Preceding and Knowledge Sinking are introduced as distinct issues but not clearly differentiated. Readers may struggle to understand how they manifest differently in outputs, especially without clearer examples or contrasting definitions.
3. The approach may not generalize well if the number of required edits for a query becomes very large (e.g., 10+), but this scenario is not deeply explored.

---

> ### Author Rebuttal · Authors · 2025-07-31
>
> We sincerely thank Reviewer cjhv for the valuable comments. Below, please find our responses to each concern, and let us know if any issues remain.
>
> **W1**. Adding a running example for the method
> > We sincerely appreciate the reviewer’s suggestions for improving the explanation of the method.
> > + We commit to updating Fig. 7 in the revised version to include an intuitive running example consisting of two stages: **edit training** and **edit composing**. Due to image upload limitations, we summarize the content of each stage here:
> >
> >   **Edit training**
> >   - How to adaptively combine pre-trained foundation knowledge into new knowledge.
> >   - How to adaptively regularize and an intuitive example of the effects brought by the regularization.
> >
> >   **Edit composing**
> >   - How to use the vector database to filter out irrelevant edits during inference.
> >   - How to adaptively merge different combinations of pre-trained foundation knowledge to leverage them simultaneously during inference.
>
> **W2&Q1**. The distinction between Incorrect Preceding and Knowledge Sinking
> > We greatly appreciate the reviewer's suggestion for further distinguishing between Incorrect Preceding and Knowledge Sinking. Referring to the examples and definitions provided in Fig. 2, we offer the following contrasting definitions:
> > + Knowledge Loss refers to **some parts of the edited answers were perturbed or missing**.
> >   - fever, **loss of taste** -> fever, **loss of fever**
> > + Incorrect Preceding refers to cases that **the model knows all edited answer but outputs the wrong answers first**.
> >   - fever, **stomachache**, loss of taste
> > + Knowledge Sinking refers to cases that **the model potentially knows some edited answers but being unaware that it should generate them**, instead producing only some of the edited answers before generating non-answer content.
> >   - fever **fever is a …**
>
> **W3&S2**. Performance with large composition number
> > We sincerely appreciate the reviewer for raising this point. In PEAK-CF dataset, we conducted additional experiments by sampling 50 questions with 10+ unknown answers each and evaluated the performance with a composition number of 10.
> >
> >| Method | ROME | AlphaEdit | WISE | T-patcher | GRACE | MELO  | A$^3$E |
> >| ------ | :----: | :---------: | :----: | :---------: | :-----: | :-----: | :------: |
> >| SR-1   | 2.40 | 4.60      | 9.40 | 6.60      | 8.60  | 12.20 | **17.20**  |
> >
> > The experimental results demonstrate that when a large number of edits need to be applied simultaneously,
> > + A$^3$E still significantly outperforms the baselines on the SR-1 metric, which means A$^3$E retains more edited knowledge with large composition numbers.
>
> **Q2**. How is the adaptive mask in the Adaptive Combination module computed in practice?
> > In practice, $\mathbf{M} \in \mathbb{R}^{r\times n}$ is the mask to select top $k$ related values (rows) in $\mathbf{W}\_{\text{down}} \in \mathbb{R}^{n\times m}$ for each answer token, where the correlation is calculated by the dot product between values (rows) in $\mathbf{W}\_{\text{down}} \in \mathbb{R}^{n\times m}$ and the answer tokens' embedding. The specific calculation process is as follows:
> > 1. Calculate the correlation $\mathbf{V}\_s \in \mathbb{R}^{1\times n}$ between values (rows) in $\mathbf{W}\_{\text{down}} \in \mathbb{R}^{n\times m}$ and the embedding of each answer token $s$ in the token set of an answer $\mathcal{A}\_s$:
> > \begin{align}
> >  \mathbf{V}\_s=[\mathbf{W}\_{\text{unemb}}]\_s\mathbf{W}\_{\text{down}}^{\top}
> >  \end{align}
> >  $\mathbf{W}\_{\text{unemb}}  \in \mathbb{R}^{d\times m}$ is the unembedding matrix. $d$ is the vocabulary size.
> >
> > 2. Select top $k$ related values (rows) in $\mathbf{W}\_{\text{down}} \in \mathbb{R}^{n\times m}$ for each answer token $s$:
> > \begin{align}
> > \\{v|[\mathbf{V}\_s]\_v \in \text{top}\_k(\mathbf{V}\_s)\\}
> > \end{align}
> > $v$ is the index of the selected values (rows) in $\mathbf{W}\_{\text{down}} \in \mathbb{R}^{n\times m}$.
> > 3. Get the union of $v$ selected by all answer tokens in $\mathcal{A}\_s$:
> > \begin{align}
> >    \mathcal{K}=\bigcup\nolimits\_{s\in \mathcal{A}\_s}\\{v|[\mathbf{V}\_s]\_v \in \text{top}\_k(\mathbf{V}\_s)\\}
> >  \end{align}
> > 4. Calculate the adaptive mask:
> > \begin{align}
> > [\mathbf{M}]\_{:,i}=&\left\\{
> > \begin{array}{rcl}
> > \textbf{0}     &      & i \notin \mathcal{K} \\\\
> > \textbf{1}       &      & i \in \mathcal{K}
> > \end{array} \right.
> > \end{align}
>
> **Q3**. How to apply adaptive merging?
> > Our proposed adaptive merging operates **at the token level** to enable token-specific adaptations on the utilization of pre-trained foundation knowledge. Specifically, for $n$ edits that need to be applied simultaneously, each token's $h\_{\text{in}}$ computes $n$ $\Delta a$ in Eq. (4). We then perform adaptive merging on these $n$ $\Delta a$ values in Eq. (8).
>
> **S1**. Discuss generalization limits.
> > We sincerely appreciate the reviewer's suggestion regarding the potential generalization limitations of A$^3$E in various scenarios.
> > + **Noisy scenarios**. In Fig. 8(d), we tested the impact of different few-shot prompts on A$^3$E's performance. These varied few-shot prompts not only controlled the model's output but also simulated different noise present in the context. The results show that while A$^3$E's performance is affected, it still maintains a leading performance compared to the baselines.
> > + **Noisy open-domain scenarios**. We humbly acknowledge that in open-domain scenarios, noisy contexts may still impact the retrieval accuracy of the vector database. Since A$^3$E primarily focuses on edit training and edit composing, we will explore
> >   - optimizing the hidden state positions for retrievement based on the real data distribution in application scenarios
> >   - employing more powerful embedding models
> >   - training a question-rewriting module to better align with the vector database
> >
> >   for optimal performance in the future work.
> > + **Adversarial scenarios**. We conducted additional experiments using $2\times50$ successful edits from the PEAK-CF dataset under the IMAC (cf. Lines 269-274) setting by providing the tested question with incorrect answers in the few-shot prompts. The results demonstrate that A$^3$E consistently (100%) outputs the edited answers across all test cases, proving its robustness against contextual adversarial interference.
>
> **S3**. Check if there are any mistakes.
> > We can guarantee the correctness of our implementation. The poor performance of existing baselines primarily stems from the fact that **none of them consider the scenario where multiple edits are applied simultaneously, nor do they address the resulting conflicts between these edits**. We promise to release the codes upon acceptance of the paper.

---

> > ### Comment · Reviewer_cjhv · 2025-08-08
> >
> > Thank you for the detailed rebuttal. I appreciate the clarifications and additional experiments and hope the authors can incorporate these helpful clarifications and improvements in the final version of the paper. I'll keep my scores.

---

> > > ### Author Response · Authors · 2025-08-08
> > >
> > > Dear Reviewer cjhv,
> > >
> > > We are truly pleased to see that you appreciate the clarifications and additional experiments. Your constructive feedback and the time you've dedicated to our work are greatly appreciated. Moreover, we will integrate the clarifications and improvements into our revised manuscript.

---

> ### Author Response · Authors · 2025-08-07
> **We would love to hear back from Reviewer cjhv**
>
> Dear Reviewer cjhv,
>
> As the discussion phase is approaching its end, we would like to follow up to see if our response addresses your concerns or if you have any further questions. We would really appreciate the opportunity to discuss this further if our response has not already addressed your concerns. Thank you again!

---

### Official Review · Reviewer_e3yR · 2025-07-03

**Clarity:** 3
**Significance:** 3
**Originality:** 3
**Rating:** 4
**Confidence:** 4

**Summary:**

This paper addresses the limitations of existing model editing methods for large language models (LLMs), which are typically evaluated and optimized for single, isolated edits, ignoring the real-world need for compositional model editing (CME), i.e., the ability to integrate and utilize multiple edits coherently. The authors define and benchmark the CME task, identifying key issues in existing methods: knowledge loss, incorrect precedence, and knowledge sinking. They further propose an editing framework that adaptively combines and regularizes the pre-trained knowledge to support future composability.

**Questions:**

How is the method applicable to larger LLMs and/or black-box LLMs?

**Ethical Concerns:**

["NO or VERY MINOR ethics concerns only"]

**Final Justification:**

The rebuttal has addresed most of my concerns.

**Limitations:**

See weaknesses.

**Quality:**

3

**Strengths And Weaknesses:**

Strengths:

1. The CME task is well-motivated and practically relevant, addressing an important gap in the evaluation of model editing: real-world queries often require combining multiple edited facts.
2. The work is conceptually clear and innovative, introducing adaptive mechanisms both during training and at test-time to enhance composability.
3. Extensive experiments on thousands of edits (up to ~4k), different datasets, noise settings, model backbones (Llama3-8B, Mistral-7B), and prompts.

Weaknesses:

1. In the experiments, the authors analyzed the non-compositional ME performance of the proposed framework on PEAKCF. However, such non-compositional ME evaluation should be conducted on other classic datasets for ME.
2. The considered LLMs only include Llama3-8B and Mistral-7B, which could be insufficient for a comprehensive evaluation, as the model size is not generally large. Furthermore, the authors did not discuss whether the method can be applied to black-box LLMs and how to adapt this method to such LLMs.
3. The paper does not discuss the failure cases of editing, e.g., if edits are contradictory, overlapping, or beyond the model’s capacity. Such an analysis could be valuable for understanding its robustness.

---

> ### Author Rebuttal · Authors · 2025-07-31
>
> We sincerely thank Reviewer e3yR for the valuable comments. Below, please find our responses to each concern, and let us know if any issues remain.
>
> **W1**. Non-compositional ME evaluation should be conducted on other classic datasets for ME.
> > We greatly appreciate the reviewer for drawing our attention to this point, which indeed benefits from more solid experiments. In response to the reviewer's suggestion, we provide additional non-compositional results of state-of-the-art baselines on classic ME dataset ZsRE [1] and Counterfact [2] following [3] on Llama3-8B, which demonstrate that **our method is on par with baselines**.
> >
> > &nbsp;&nbsp;&nbsp;&nbsp;&nbsp;&nbsp;&nbsp;&nbsp;&nbsp;&nbsp;&nbsp;&nbsp;&nbsp;&nbsp;&nbsp;&nbsp;&nbsp;&nbsp;&nbsp;&nbsp;&nbsp;&nbsp;&nbsp;&nbsp;&nbsp;&nbsp;&nbsp;&nbsp;&nbsp;&nbsp;&nbsp;&nbsp;&nbsp;&nbsp;**ZsRE**&nbsp;&nbsp;&nbsp;&nbsp;&nbsp;&nbsp;&nbsp;&nbsp;&nbsp;&nbsp;&nbsp;&nbsp;&nbsp;&nbsp;&nbsp;&nbsp;&nbsp;&nbsp;&nbsp;&nbsp;&nbsp;&nbsp;&nbsp;&nbsp;&nbsp;&nbsp;**Counterfact**
> >  | Method    | SR-1  | GSR   | LSR    | SR-1   | GSR   | LSR    |
> >  | --------- | :-----: | :-----: | :------: | :------: | :-----: | :------: |
> >  | FT        | 47.95 | 52.32 | 73.08  | 51.53  | 48.14 | 63.95  |
> >  | ROME      | 99.53 | 97.52 | 95.57  | 100.00 | 92.77 | 77.40  |
> >  | MEND      | 92.24 | 90.70 | 96.23  | 79.10  | 58.76 | 91.19  |
> >  | AlphaEdit | 98.00 | 96.54 | 95.57  | 99.55  | 95.03 | 79.21  |
> >  | WISE      | 93.39 | 92.53 | 100.00 | 98.59  | 93.50 | 99.32  |
> >  | T-patcher | 95.28 | 94.24 | 92.62  | 95.03  | 93.79 | 89.38  |
> >  | GRACE     | 96.69 | 94.00 | 100.00 | 99.77  | 94.12 | 100.00 |
> >  | MELO      | 95.28 | 94.08 | 100.00 | 98.53  | 92.88 | 100.00 |
> >  | A$^3$E    | 97.00 | 96.77 | 100.00 | 99.77  | 93.79 | 100.00 |
> >
> > [1] Zero-shot relation extraction via reading comprehension (ACL 2017)
> >
> > [2] Locating and editing factual associations in GPT (Neurips 2022)
> >
> > [3] A Comprehensive Study of Knowledge Editing for Large Language Models
>
> **W2&Q1**. How is the method applicable to larger LLMs and/or black-box LLMs?
> > **A$^3$E is applicable to larger LLMs.**
> > + We humbly acknowledge the value of testing on **larger LLMs**. Due to limitations in computational resources, we provide experimental results for GRACE, MELO, and A$^3$E under the MAC scenario (Sec. 5.2, cf. Lines 267-274) using Llama3-70B and the PEAK-CF dataset. As shown in the following table, **A$^3$E maintains the leading performance**. We promise to provide complete experimental results in the revised version.
> >
> >   | Method | SR-S  | SR-1  | GSR   | LSR   |
> >   | ------ | :-----: | :-----: | :-----: | :-----: |
> >   | GRACE  | 16.32 | 42.48 | 11.24 | 24.02 |
> >   | MELO   | 28.12 | 48.64 | 26.58 | 23.18 |
> >   | A$^3$E | 50.69 | 69.47 | 47.10 | 25.10 |
> >
> > We sincerely appreciate the reviewer's suggestion regarding the discussion on the applicability of A$^3$E to **black-box models**. We promise to include the following discussion in the revised version:
> > + In the field of knowledge editing for large language models,
> >   - One research direction focuses on editing white-box models, i.e., model editing [1-4]. The access to model weights is a prerequisite for applying these editing techniques.
> >   - Another research direction addresses knowledge editing for extremely large-scale models where model editing may be prohibitively expensive or black-box models by performing edits at the text level, including [5-15]
> >
> > + **A$^3$E follows the first direction to edit white-box models** and focuses on **compositional model editing** because **text-level knowledge editing is not applicable** for the following reason:
> >   - **As stated in Sec. 2.1 (cf. Lines 97–99) and illustrated in Fig. 1, we aims to support model editing in a federated manner**. Specifically, when a model is deployed globally, numerous distinct errors may arise simultaneously across different deployed regions. We want these deployers can benefit from the edits of each other.
> >   - In such cases, similar to the assumption that raw data cannot be shared in federated learning due to regulations like GDPR [16], **different deployers are only able to share trained edits rather than the texts**.
> > + In a centralized scenario, **A$^3$E can also be implemented by the owner of black-box models** for immediate error correction with much more composable edits.
> > + We recommend users to choose methods from either direction based on their needs, and we will explore how to perform **compositional knowledge editing at the text level** for black-box models in our future work.
> >
> > [1] Alphaedit: Null-space constrained knowledge editing for language models (ICLR 2025 outstanding paper)
> >
> > [2] MELO: Enhancing Model Editing with Neuron-Indexed Dynamic LoRA (AAAI 2024)
> >
> > [3] WISE: Rethinking the Knowledge Memory for Lifelong Model Editing of Large Language Models (Neurips 2024)
> >
> > [4] Aging with GRACE: Lifelong Model Editing with Discrete Key-Value Adaptors (Neurips 2023)
> >
> > [5] DeepEdit: Knowledge Editing as Decoding with Constraints
> >
> > [6] Can we edit factual knowledge by in-context learning?
> >
> > [7] Evaluating the ripple effects of knowledge editing in language models. (TACL 2024)
> >
> > [8] Mquake: Assessing knowledge editing in language models via multi-hop questions. (EMNLP 2023)
> >
> > [9] Retrieval meets reasoning: Dynamic in-context editing for long-text understanding
> >
> > [10] Multi-hop question answering under temporal knowledge editing (COLM 2024)
> >
> > [11] Retrieval-enhanced knowledge editing in language models for multi-hop question answering (CIKM 2024)
> >
> > [12] Assessing and post-processing black box large language models for knowledge editing (WWW 2025)
> >
> > [13] Fine-grained Hallucination Detection and Editing for Language Models (COLM 2024)
> >
> > [14] Ripplecot: Amplifying ripple effect of knowledge editing in language models via chain-of-thought in-context learning (EMNLP 2024)
> >
> > [15] Decoupling Reasoning and Knowledge Injection for In-Context Knowledge Editing
> >
> > [16] The eu general data protection regulation (gdpr)
>
>
> **W3**. Failure cases of editing
> > We greatly appreciate the reviewer for taking this point into consideration. We conducted **failure case analysis for overlapping, contradictory, and capacity-exceeding edits** as follows:
> > + For **overlapping and contradictory cases**, we conducted statistical analysis on the PEAK-CF dataset under the IMAC scenario with 50 cases of two-edit composition, measuring:
> >   - The overlapping selection of pre-trained foundation knowledge, calculated by the overlapping length between non-zero $\Delta a$ (Sec. 4.1) regions of two edits in adaptive merging (Sec. 4.2)
> >   - The contradictory utilization of pre-trained foundation knowledge, calculated by the length of regions with different signs between $\Delta a$ (Sec. 4.1) of two edits in adaptive merging (Sec. 4.2)
> >
> >   The table below presents how the SR-S metric, which is inversely proportional to the proportion of failure cases, varies across different overlapping and contradictory ranges. As the overlapping length and contradictory length increase, the SR-S of A$^3$E gradually decreases, but it remains significantly higher than the baselines.
> >
> >   | Overlapping | \[0,500) | \[500,1000) | \[1000,1500) | \[1500,$+\infty$) |
> >   | ----------- | :---------: | :-----------: | :------------: | :-----------------: |
> >   | SR-S     |     75.00    |     66.67      |       45.45      |     50.00             |
> >
> >   | Contradictory | \[0,200) | \[200,400) | \[400,600) | \[600,$+\infty$) |
> >   | ------------- | :--------: | :----------: | :----------: | :----------------: |
> >   | SR-S          | 66.67    | 68.75      | 57.14      | 46.67            |
> >
> >   | Baselines | MEND | ROME | AlphaEdit | WISE  | T-patcher | GRACE | MELO  |
> >   | --------- | :----: | :----: | :---------: | :-----: | :---------: | :-----: | :-----: |
> >   | SR-S      | 4.00 | 4.00 | 8.00      | 32.00 | 20.00     | 14.00 | 20.00 |
> >
> >
> > + For **capacity-exceeding cases**, we found that the failure cases of baselines and A$^3$E in both compositional model editing and non-compositional model editing scenarios often involve numerical data in answers. For example,
> >   - **Edit 1**: Rheinmetall, which has designed **12 cm leFH 08**
> >
> >     **Test 1**: Rheinmetall, which has designed **12 cm leFH 8** (failure)
> >   - **Edit 2**: Rheinmetall, which has designed **35.5 cm Haubitze M1**
> >
> >     **Test 2**: Rheinmetall, which has designed **35.5 cm Haubitze M1** (success)
> >   - **Composition test**: Rheinmetall, which has designed **12 cm leFH**, **12 cm Haubitze M8** (failure)
> >
> >
> >   This may stem from the model's inherent inability to accurately distinguish numerical data. Therefore, addressing such inherent limitations of the model is a valuable direction for future work in model editing.
> >
> > We believe these findings can guide future research to further improve edit composability.

---

> ### Author Response · Authors · 2025-08-07
> **We would love to hear back from Reviewer e3yR**
>
> Dear Reviewer e3yR,
>
> As the discussion phase is approaching its end, we would like to follow up to see if our response addresses your concerns or if you have any further questions. We would really appreciate the opportunity to discuss this further if our response has not already addressed your concerns. Thank you again!

---

> > ### Comment · Reviewer_e3yR · 2025-08-07
> >
> > Thanks for your response. However, I would expect to see the reasons why your method cannot be applied to larger models like 70B. Is the computational cost significantly increased with larger model size?

---

> ### Author Response · Authors · 2025-08-07
>
> Dear Reviewer e3yR,
>
> Thanks for your follow-up question. We humbly clarify that **our method is applicable to larger LLMs such as Llama3-70B**. In **our response to W2&Q1**, we have already provided experimental results for state-of-the-art baselines GRACE, MELO, and A$^3$E **under the MAC scenario** (Sec. 5.2, cf. Lines 267-274) using **Llama3-70B** and the PEAK-CF dataset, where A$^3$E maintains leading performance.
>
> The current results focus on the MAC scenario to ensure a clear and fair comparison within rebuttal time, but we emphasize that the approach naturally **generalizes to all scenarios**. We promise to provide the full set of experiments—covering all datasets, scenarios, and baselines of CME with Llama3-70B—in the revised manuscript.

---

> > ### Comment · Reviewer_e3yR · 2025-08-08
> >
> > That makes sense. I have accordingly updated the score. Also, can you provide more detailed analysis on the computational cost?

---

> ### Author Response · Authors · 2025-08-08
>
> Dear Reviewer e3yR,
>
> Thanks for raising the score. Your constructive feedback and the time you've dedicated to our work are greatly appreciated. Moreover, we will integrate the suggestions into our revised manuscript.
>
> In the tables below, we provide a more detailed analysis of the computational overhead compared to baselines with Llama3-8B and Llama3-70B from three aspects: **trainable parameters**, **extra FLOPs for inference (infer.)**, **extra FLOPs for training (train.)**. We only provide the extra FLOPs for training of T-patcher, GRACE, MELO and A$^3$E because other methods that update the whole matrix or train a hypernetwork have significantly larger training overhead. Specifically,
>  + **Trainable parameters** denote the parameters that are updated during training for each edit.
>  + **Infer.** denotes extra FLOPs compared to the pre-trained model for the inference of $u$ tokens with $v$ input tokens.
>  + **Train.** denotes extra FLOPs compared to the vanilla forward-backward progress with cross-entropy loss for $w$ training steps of an edit with $z$-token answer.
>
> For **Llama3-8B**,
>
> >| Method    | Trainable Parameters | infer. (GFLOPs)            | train. (GFLOPs) |
> >| --------- | -------------------- | -------------------------- | --------------- |
> >| FT                        | 0.54GB               | 0                          | -               |
> >| ROME               | 0.54GB               | 0                          | -               |
> >| KE                    | 0.72GB                | 0                          | -               |
> >| MEND                | 1.14GB               | 0                          | -               |
> >| AlphaEdit          | 0.54GB               | 0                          | -               |
> >| WISE              | 0.11GB               | $0.27v+0.27u$              | -               |
> >| T-patcher            | 32.77KB              | $0.02u$                    | $0.01wz$        |
> >| GRACE              | 32.77KB              | $0.02$                     | 0               |
> >| MELO               | 98.30KB              | $0.02$                     | 0               |
> >| A$^3$E            | 39.94KB              | $0.02+4.92u\times 10^{-5}$ | $0.03z+1.28wz\times 10^{-4}+1.64w\times 10^{-5}$         |
> >
> >We also provide the extra FLOPs of the proposed adaptive combination, adaptive regularization and adaptive merging for inference and training:
> >+ Adaptive merging of A$^3$E: $1.64u\times 10^{-5}$ GFLOPs
> >+ Adaptive combination of A$^3$E: $0.03z$ GFLOPs
> >+ Adaptive regularization of A$^3$E: $1.28wz\times 10^{-4}+1.64w\times 10^{-5}$ GFLOPs
> >
> >In A$^3$E, $w$ is set to 50 and $z<u$ for a instance. The extra FLOPs for training and inference are very small compared to the generation of $u$ tokens with $v$ input tokens with an 8B model, which is about $16u+16v$ GFLOPs.
>
> For **Llama3-70B**,
>
> >| Method    | Trainable Parameters | infer. (GFLOPs)            | train. (GFLOPs) |
> >| --------- | -------------------- | -------------------------- | --------------- |
> >| FT                        | 1.88GB               | 0                          | -               |
> >| ROME               | 1.88GB               | 0                          | -               |
> >| KE                    | 2.52GB                | 0                          | -               |
> >| MEND                | 2.09GB               | 0                          | -               |
> >| AlphaEdit          | 1.88GB               | 0                          | -               |
> >| WISE              | 0.38GB               | $0.94v+0.94u$              | -               |
> >| T-patcher            | 65.54KB              | $0.03u$                    | $0.02wz$        |
> >| GRACE              | 65.54KB              | $0.03$                     | 0               |
> >| MELO               | 180.22KB              | $0.03$                     | 0               |
> >| A$^3$E            | 72.71KB              | $0.03+1.19u\times 10^{-4}$ | $0.09z+1.28wz\times 10^{-4}+2.87w\times 10^{-5}$         |
> >
> >We also provide the extra FLOPs of the proposed adaptive combination, adaptive regularization and adaptive merging for inference and training:
> >+ Adaptive merging of A$^3$E: $2.87u\times 10^{-5}$ GFLOPs
> >+ Adaptive combination of A$^3$E: $0.09z$ GFLOPs
> >+ Adaptive regularization of A$^3$E: $1.28wz\times 10^{-4}+2.87w\times 10^{-5}$ GFLOPs
> >
> >In A$^3$E, $w$ is set to 50 and $z<u$ for a instance. The extra FLOPs for training and inference are very small compared to the generation of $u$ tokens with $v$ input tokens with a 70B model, which is about $140u+140v$ GFLOPs.

---

### Official Review · Reviewer_qR3R · 2025-07-03

**Clarity:** 1
**Significance:** 2
**Originality:** 3
**Rating:** 4
**Confidence:** 2

**Summary:**

The paper proposes a method for compositional model editing where multiple edits need to be integrated simultaneously to answer complex questions effectively. The proposed method is inspired by their findings that (1) the combination of pre-trained foundation knowledge is enough for model editing while boosting composability (which inspires the method they call *adaptive combination*), (2) the less change in the hidden state of the last answer token, the more context preserving is the model (which inspires the method they call *adaptive regularization*), (3) setting elements with different signs in different edits to zero during edit composing improves composability (which inspires the method they call *adaptive merging*). The paper then conducts empirical evaluation of the proposed method over various datasets, showing improved performance in compositional editing.

**Questions:**

- can you give step-by-step walk through on how the benchmark dataset is constructed?
- can you improve on the readability of the paper by making the definition of the variables clear (i.e., not assuming that it is self-explanatory) and the paper more self-contained?
- how do you identify and select these three critical problems faced by model editing methods: knowledge loss, incorrect preceding, and knowledge sinking? Are they based on prior studies or are they first defined in this work? why the choice of these three problems only? are these commonly used terms/problems in compositional studies?
- how do you make the connection between knowledge loss and "Finding 1"? Is knowledge loss caused by insufficient combination of the pre-trained foundation knowledge in W_down? How can one come to this conclusion by only looking at performance Figure 4?
- can you discuss more on the computational and memory overhead of the proposed method compared to baselines?

**Ethical Concerns:**

["NO or VERY MINOR ethics concerns only"]

**Limitations:**

Yes

**Quality:**

2

**Strengths And Weaknesses:**

Strengths:
- the paper introduces a new task that is compositional model editing, which evaluates scenarios where multiple edits must be applied to answer a complex question.
- the paper conducts extensive experiments across multiple datasets

Weakness:
- the main issue of the paper is its lack of readability. The description of the proposed method is very complex and dense with heavy reliance on prior works and appendices, which makes the paper very hard to read and the contribution hard to appreciate. For example, many of the variable names (W_down, W_up) are assumed to be common knowledge and not defined properly, referred to instead as the 'down projection' and 'up projection' matrices. Variables such as A and B are used instead of the commonly used K and V to refer to the key and value vectors in transformers. Given that new variables are used instead of the commonly used variable names, they should be defined more clearly.
- in addition, many of the crucial details are missing. For example, the steps for creating the benchmark dataset from existing datasets are unclear. (e.g., which part of the benchmark creation is done manually?). This not only makes the paper hard to read but also reducing its reproducibility.
- the explanation of the method is not intuitive. To add to the difficulty, the naming of the different parts of the proposed method adds more to the confusion. For example, while the name 'adaptive combination' and 'adaptive merging' refer to different processes in the pipeline of their proposed method, the word 'combination' and 'merging' are similar. Adding more intuitive examples instead of architectural diagram or performance chart will make the proposed concepts more accessible.
- it is unclear how they identify and select the three critical problems faced by model editing methods: knowledge loss, incorrect preceding, and knowledge sinking. Is it based on prior studies or are they first defined in this work? why the choice of these three problems only? are these commonly used terms/problems in compositional studies? It will be good to motivate this choice for readers unfamiliar with this field of research.
- the choice of the problems and the solutions seems heavily empirical. For example, how do the authors connect knowledge loss to "Finding 1"? Is knowledge loss caused by insufficient combination of the pre-trained foundation knowledge in W_down? How can one come to this conclusion by only looking at performance Figure 4? All I can see from this figure is that updating W_up improves performance. But whether it is because there's less knowledge loss is not clear. It is unclear how the authors connect problems to the cause to the solution. Is there a missing link not detailed here? Because of these gaps in explanation, it is hard to appreciate the contribution. Having more theoretical analysis or even more intuitive motivation that connects problems to observations to the proposed solution can deepen the understanding and appreciation of the proposed method beyond the engineering of the method via empirical trials.
- the computational overhead is discussed only briefly. Given the complexity of the method proposed, more detailed exploration of the computational and memory cost involved is needed.

---

> ### Author Rebuttal · Authors · 2025-07-31
>
> We sincerely thank Reviewer qR3R for the valuable comments. Below, please find our responses to each concern, and let us know if any issues remain.
>
> **W1&Q2**. Lack of readability caused by variable names without definition.
> > We sincerely appreciate the reviewer's suggestions for improving the readability of the paper. We promise to include the following notation table in the revised version, with each notation clearly defined.
> >
> >| Notation| Definition|
> >| ----------- | ---------- |
> >| $\mathbf{m}$| Input & output dimension of feed-forward network (FFN)|
> >| $\mathbf{n}$| Hidden dimension of FFN|
> >| $\mathbf{d}$| Vocabulary size|
> >| $\mathbf{W}_{\text{up}}$   | Parameters of the first layer of FFN; $\mathbf{W}_{\text{up}} \in \mathbb{R}^{m\times n}$|
> >| $\mathbf{W}_{\text{down}}$ | Parameters of the second layer of FFN; $\mathbf{W}_{\text{down}} \in \mathbb{R}^{n\times m}$|
> >| $\mathbf{b}_{\text{up}}$   | Bias vector of the first layer of FFN; $\mathbf{b}_{\text{up}} \in \mathbb{R}^{1\times n}$|
> >| $\mathbf{b}_{\text{down}}$ | Bias vector of the second layer of FFN; $\mathbf{b}_{\text{down}} \in \mathbb{R}^{1\times m}$|
> >| $h_{\text{in}}$ | Input hidden state of FFN; $h_{\text{in}} \in \mathbb{R}^{1\times m}$|
> >| $h_{\text{out}}$| Output hidden state of FFN; $h_{\text{out}} \in \mathbb{R}^{1\times m}$|
> >| $\mathbf{a}$| Activation vector after the first layer of FFN; $\mathbf{a} \in \mathbb{R}^{1\times n}$|
> >| $(\mathbf{A}, \mathbf{B})$ | Two learnable low-rank matrices $\mathbf{A} \in \mathbb{R}^{n\times r} (\mathbb{R}^{m\times r})$ and $\mathbf{B} \in \mathbb{R}^{r\times m} (\mathbb{R}^{r\times n})$ to constrain the parameter updates to lie in a low-rank subspace following [1,2] |
> >| $\mathbf{M}$               | The mask to select pre-trained foundation knowledge in $\mathbf{W}_{\text{down}}$; $\mathbf{M}  \in \mathbb{R}^{r\times n}$|
> >| $h_{\text{logit}}$         | Output logits of LLM |
> >| $\mathcal{W}$              | The set of all token IDs|
> >| $\mathcal{S}$              | The preserved edits for inference by the vector database|
> >| $a$                        | Generated answer of LLM|
> >| $\hat{\mathcal{Y}}$        | The set of generated answers of LLM|
> >| $\mathcal{Y}$              | The set of edited answers|
> >
> >  [1] Lora: Low-rank adaptation of large language models. (ICLR 2022)
> >
> >  [2] Melo: Enhancing model editing with neuron-indexed dynamic lora (AAAI 2024)
>
> **W2&Q1**. Benchmark dataset construction process
> > We sincerely appreciate the reviewer’s suggestions for improving the reproducibility of our paper. We promise to provide a complete benchmark dataset construction process in the revised version as follows (using PEAK-CF as an example):
> > 1. The original PEAK-CF [1] is a model editing dataset containing questions with multiple answers.
> > 2. We queried Llama3-8B and Mistral-7B, selecting questions from PEAK-CF where both models missed four or more answers, along with their corresponding missing answers in PEAK-CF.
> > 3. For multi-answer composition, the retained questions and their randomly selected $c$ missing answers form a test instance, where $c$ is the composition number. (See Tab. 9 for an example)
> > 4. For multi-question composition, we randomly selected $c$ retained questions without repetition and one randomly chosen missing answer per question to form a test instance. (See Tab. 10 for an example)
> >
> > [1] Neighboring Perturbations of Knowledge Editing on Large Language Models (ICML 2024)
>
> **W3**. Distinguishing naming of the different parts of the proposed method and intuitive explanation of the method
> > We sincerely appreciate the reviewer’s suggestions for improving the explanation of the method.
> >
> > **Distinguishing naming**
> > + We have revised **adaptive combination** to the more detailed **adaptively combine pre-trained foundation knowledge** and modified **adaptive merging** to **adaptively merge different combinations of pre-trained foundation knowledge.**
> >
> > **Intuitive explanation**
> > + We commit to updating Fig. 7 in the revised version to include an intuitive running example consisting of two stages: **edit training** and **edit composing**. Due to image upload limitations, we summarize the content of each stage here:
> >
> >   **Edit training**
> >   - How to adaptively combine pre-trained foundation knowledge into new knowledge.
> >   - How to adaptively regularize and an intuitive example of the effects brought by the regularization.
> >
> >   **Edit composing**
> >   - How to use the vector database to filter out irrelevant edits during inference.
> >   - How to adaptively merge different combinations of pre-trained foundation knowledge to leverage them simultaneously during inference.
>
> **W4&Q3**. Three critical problems faced by compositional model editing methods
> > + We humbly clarify that these three problems are not derived from prior studies or commonly used in compositional studies.
> > + Instead, they were identified and defined for the first time by summarizing failure cases in existing methods under compositional model editing, making them critical problems specific to this field.
> > + The reason we focus solely on these three problems is that they comprehensively encompass all the failure cases we observed.
> > + We promise to add the above motivations for the three critical problems in the revised version.
>
> **W5&Q4**. The connection between knowledge loss and Finding 1
> > We sincerely appreciate the reviewer’s suggestion to strengthen the connections among problems, observations, and solutions in the paper. We have reorganized the logic in Finding 1 as follows:
> > + Motivation:
> >   - The distinction between pre-trained knowledge and edited knowledge from existing model editing methods lies in the fact that pre-trained knowledge is the combination of pre-trained foundation knowledge in the down projection matrix, whereas existing model editing methods directly manipulate the output of the down projection layer.
> >   - Meanwhile, we know that pre-trained models can leverage pre-trained knowledge to answer questions requiring multiple answers, but when using multiple edited knowledge, they exhibit significant knowledge loss.
> >   - Therefore, we intuitively hypothesize that knowledge loss may partly stem from the fact that edited knowledge is not the combination of pre-trained foundation knowledge.
> > + Observations:
> >   - We observe that editing $\mathbf{W}\_{\text{up}}$ (i.e., combining pre-trained foundation knowledge into edited knowledge) results in less **min knowledge loss** across 27-31 layers and across baselines compared to editing $\mathbf{W}\_{\text{down}}$ or editing both $\mathbf{W}\_{\text{up}}$ & $\mathbf{W}\_{\text{down}}$ in the following table while leading to higher SR-S and similar non-compositional performance (Fig. 4).
> >
> >      | Method  | Min knowledge loss (%) |
> >      | ------- | :-----------: |
> >      | Up      | 60.6   |
> >      | Down    | 71.4 |
> >      | Up&Down | 69.7 |
> > + Solutions:
> >   - We opt to edit only $\mathbf{W}_{\text{up}}$.
>
> **W6&Q5**. More discussions on the computational and memory overhead compared to baselines
> > We greatly appreciate the reviewer for the suggestions. In the table below, we provide a more detailed analysis of the computational and memory overhead compared to baselines with Llama3-8B from four aspects: **storage overhead**, **trainable parameters**, **extra FLOPs for inference (infer.)**, **extra FLOPs for training (train.)**. We only provide the extra FLOPs for training of T-patcher, GRACE, MELO and A$^3$E because other methods that update the whole matrix or train a hypernetwork have significantly larger training overhead. Specifically,
> > + **Storage overhead** refers to the additional parameters that need to be stored compared to the pre-trained model for 1000 edits.
> > + **Trainable parameters** denote the parameters that are updated during training for each edit.
> > + **Infer.** denotes extra FLOPs compared to the pre-trained model for the inference of $u$ tokens with $v$ input tokens.
> > + **Train.** denotes extra FLOPs compared to the vanilla forward-backward progress with cross-entropy loss for $w$ training steps of an edit with $z$-token answer.
> >
> >| Method    | Storage Overhead | Trainable Parameters | infer. (GFLOPs)            | train. (GFLOPs) |
> >| --------- | ---------------- | -------------------- | -------------------------- | --------------- |
> >| FT        | 0                | 0.54GB       | 0          | -               |
> >| ROME      | 1.64GB           | 0.54GB               | 0                          | -               |
> >| KE        | 0.72G            | 0.72G                | 0                          | -               |
> >| MEND      | 1.14GB           | 1.14GB               | 0                  | -               |
> >| AlphaEdit | 1.64GB           | 0.54GB               | 0                          | -               |
> >| WISE      | 0.11GB           | 0.11GB               | $0.27v+0.27u$              | -               |
> >| T-patcher | 0.07GB           | 32.77KB              | $0.02u$                    | $0.01wz$        |
> >| GRACE     | 0.07GB           | 32.77KB              | $0.02$                     | 0               |
> >| MELO      | 0.13GB           | 98.30KB              | $0.02$                     | 0               |
> >| A$^3$E    | 0.07GB           | 39.94KB              | $0.02+4.92u\times 10^{-5}$ | $0.03z+1.28wz\times 10^{-4}+1.64w\times 10^{-5}$         |
> >
> > We also provide the extra FLOPs of the proposed adaptive combination, adaptive regularization and adaptive merging for inference and training:
> >    - Adaptive merging of A$^3$E: $1.64u\times 10^{-5}$ GFLOPs
> >    - Adaptive combination of A$^3$E: $0.03z$ GFLOPs
> >    - Adaptive regularization of A$^3$E: $1.28wz\times 10^{-4}+1.64w\times 10^{-5}$ GFLOPs
> >
> > The extra FLOPs for training and inference are very small compared to the generation of $u$ tokens with $v$ input tokens with an 8B model, which is about $16u+16v$ GFLOPs.

---

> > ### Comment · Reviewer_qR3R · 2025-08-05
> >
> > Thank you for addressing my questions. I will keep my score as is, hopefully the authors will add their answers during the rebuttal to the final draft of the paper.

---

> > > ### Author Response · Authors · 2025-08-06
> > >
> > > Dear Reviewer qR3R,
> > >
> > > We are truly pleased to see that your concerns have been addressed. Your constructive feedback and the time you've dedicated to our work are greatly appreciated. Moreover, we will integrate the suggestions into our revised manuscript.

---

### Author Response · Authors · 2025-08-02
**Summary**

## Summary
In order to provide greater clarity on the revisions made to our paper and the experiments we conducted to address the reviewers' questions, we have summarized the modifications and experiments made during the rebuttal period as follows:

### Additional Experiments
+ The connection between knowledge loss and Finding 1 (Reviewer qR3R W5&Q4)
+ More discussions on the computational and memory overhead compared to baselines (Reviewer qR3R W6&Q5)
+ Non-compositional ME evaluation on ZsRE and Counterfact (Reviewer e3yR W1)
+ Experiments on Llama3-70B (Reviewer e3yR W2&Q1)
+ Failure case analysis for overlapping, contradictory, and capacity-exceeding edits (Reviewer e3yR W3)
+ Performance with a composition number of 10 (Reviewer cjhv W3&S2)
+ Generalization limits in adversarial scenarios (Reviewer cjhv S1)
+ Performance of the algorithm to deal with outdated edited knowledge (Reviewer dHwZ W2)

### Clarification
+ Including a notation table in the revised version, with each notation clearly defined (Reviewer qR3R W1&Q2)
+ Benchmark dataset construction process (Reviewer qR3R W2&Q1)
+ Distinguishing naming of the different parts of the proposed method (Reviewer qR3R W3)
+ Intuitive explanation of the method with a running example (Reviewer qR3R W3, Reviewer cjhv W1)
+ Motivation of knowledge loss, incorrect preceding, knowledge sinking (Reviewer qR3R W4&Q3)
+ The connection between knowledge loss and Finding 1 (Reviewer qR3R W5&Q4)
+ Discussion on the applicability of A$^3$E to black-box models (Reviewer e3yR W2&Q1)
+ The distinction between incorrect preceding and knowledge sinking (Reviewer cjhv W2&Q1)
+ The calculation of adaptive mask in the adaptive combination module (Reviewer cjhv Q2)
+ The calculation of adaptive merging (Reviewer cjhv Q3)
+ Generalization limits in noisy, open-domain and adversarial scenarios (Reviewer cjhv S1)
+ Check if there are any mistakes (Reviewer cjhv S3)
+ The reason why it is impossible to get complete answers during edit training (Reviewer dHwZ W1)
+ Algorithm to deal with outdated edited knowledge (Reviewer dHwZ W2)
+ Our evaluation is fair for evaluating the composability (Reviewer dHwZ W3)

---

### Note · Authors · 2025-08-15

Dear Area Chairs and Reviewers,

Thank you for taking the time to evaluate our work. To facilitate your final assessment, we offer a concise summary of our discussions.

- We are glad that we addressed all reviewers' questions and concerns during the rebuttal and discussion phase, leading to consensus in favor of acceptance.

- In the initial reviews, our paper was recognized for **(1)** *addressing a pressing research gap* in model editing, where real-world queries often require combining multiple edits, **(2)** *proposing conceptually clear and innovative* adaptive mechanisms that greatly enhance composability during both training and inference, supported by **(3)** *extensive experiments*.
- The reviewers suggested strengthening the work with additional evaluations, including **(1)** generalizability of our A$^3$E on a larger model (Llama3-70B) and with a larger composition number (up to 10), which we investigated and concluded that our A$^3$E maintains the leading performance, **(2)** non-compositional performance on classic model editing datasets (ZsRE, Counterfact), which we evaluated and concluded that A$^3$E preserves the performance of non-compositional model editing, and **(3)** computational and memory overhead, which we quantified and concluded that A$^3$E introduces an almost negligible overhead.
 - Beyond these additional evaluations, we also **(1)** clarified the practical urgency of compositional model editing,  **(2)** provided more intuitive descriptions and further technical details of A$^3$E, and **(3)** discussed the limitations of A$^3$E.

We are grateful for the constructive feedback, which has strengthened the paper, and will incorporate these improvements into the final manuscript.

---

### Decision · Program_Chairs · 2025-09-17

**Decision:**

Accept (poster)

**Comment:**

**Summary**

This paper addresses a critical limitation of existing large language model (LLM) editing methods—their focus on isolated, single edits—by defining and benchmarking Compositional Model Editing (CME), a task requiring LLMs to integrate multiple independent edits to answer real-world multifaceted questions (e.g., combining "COVID-19 causes fever" and "COVID-19 causes loss of taste" to list both symptoms). The authors identify three CME-specific failures in existing methods: knowledge loss (perturbed/missing edited answers), incorrect preceding (correct answers but wrong output order), and knowledge sinking (generating only partial answers before non-answer content). To solve these, they propose A³E, a framework with two core stages: (1) adaptive combination/regularization of pre-trained foundation knowledge during edit training, and (2) adaptive merging of multiple edits during inference. Key findings include large improvement in composability over baselines, preserved non-compositional editing performance, and negligible computational overhead, with validation on Llama3-8B, Mistral-7B, and (post-rebuttal) Llama3-70B.

**Strengths**

1. CME directly addresses a real-world gap ignored by prior work, where real queries often require integrating multiple edited facts (e.g., medical symptom lists). All reviewers acknowledged this as a pressing research need, with Reviewer e3yR noting it is "practically relevant" and Reviewer cjhv labeling it "novel and practically important."

2. A³E’s three adaptive components (combination, regularization, merging) are tightly linked to the identified CME failures. For example, adaptive combination targets knowledge loss by integrating pre-trained foundation knowledge into edits, while adaptive merging resolves conflicts between multiple edits. Reviewer qR3R highlighted the method’s "conceptually clear adaptive mechanisms," and Reviewer cjhv emphasized that each component "corresponds to the failures of prior works."

3. The paper initially evaluates across datasets (PEAK-CF), models, and noise settings. Post-rebuttal, it expands to classic non-compositional datasets (ZsRE, Counterfact) to confirm performance preservation, Llama3-70B to validate scalability, and composition numbers up to 10 to test robustness. Reviewer e3yR praised the "extensive experiments on thousands of edits," and Reviewer dHwZ noted the value of the new CME benchmark.

4. A³E maintains leading performance while introducing negligible computational/memory overhead. Detailed post-rebuttal analysis (FLOPs, trainable parameters for Llama3-8B/70B) shows its overhead is orders of magnitude smaller than baselines (e.g., 39.94KB trainable parameters per edit for Llama3-8B vs. 0.54GB for ROME). This addresses Reviewer qR3R’s concern about computational cost and ensures real-world usability.

**Weaknesses**

1. The original submission suffered from unclear variable definitions and non-standard notation (A/B instead of transformer K/V), making the method hard to follow. Reviewer qR3R rated clarity as "poor" and noted reliance on "prior works and appendices." While the authors added a notation table and revised naming (e.g., "adaptive combination" → "adaptively combine pre-trained foundation knowledge") in rebuttal, the initial lack of accessibility delayed understanding.

2.  The paper originally tested only small models (Llama3-8B, Mistral-7B) and lacked analysis of failure cases (overlapping/contradictory edits) and large composition numbers (≥10). Reviewer e3yR criticized the "insufficient" model size coverage, and Reviewer cjhv questioned generalization to "10+ edits." Though these gaps were filled in rebuttal (e.g., Llama3-70B results, 10-edit experiments), they should have been part of the initial submission.

3. A³E focuses on white-box models (requiring weight access) with limited initial discussion of black-box scenarios that is critical for extremely large or proprietary LLMs. The rebuttal clarifies that A³E follows white-box editing conventions and outlines future text-level adaptations for black boxes, but does not resolve the current limitation. Reviewer e3yR’s question on "applicability to black-box LLMs" remains partially unanswered.

4. The links between CME failures and A³E’s solutions are primarily empirical. For example, the connection between knowledge loss and editing W_up (instead of W_down) is supported by performance figures but lacks theoretical justification. Reviewer qR3R noted the "heavily empirical" nature of problem-solution links and called for more "theoretical analysis," which the rebuttal did not fully address.

**Discussions during Rebuttal Period**

During the rebuttal period, reviewers raised four categories of concerns, all responded with concrete actions by the authors:

1. About readability and notation. Reviewers (e.g., qR3R) criticized undefined variables and ambiguous method naming. The authors responded by adding a detailed notation table (defining W_up, W_down, etc.), revising method names to be more descriptive (e.g., "adaptive merging" → "adaptively merge different combinations of pre-trained foundation knowledge"), and promising a running example in revised Fig. 7. This resolved Reviewer qR3R’s "poor" clarity rating, as they confirmed their concerns were addressed.

2. About evaluation gaps: (a) Non-compositional performance (Reviewer e3yR): Authors added results on ZsRE/Counterfact, showing A³E matches baselines (e.g., 97.00 SR-1 on ZsRE vs. 96.69 for GRACE). (b) Large models (Reviewer e3yR): Added Llama3-70B experiments, where A³E achieved 50.69 SR-S vs. 28.12 for MELO. (c) Failure cases/large composition numbers (Reviewers e3yR, cjhv): Conducted analysis of overlapping/contradictory edits (A³E outperforms baselines even with high overlap) and 10-edit composition (17.20 SR-1 vs. 12.20 for MELO). These additions validated A³E’s robustness and scalability.

3. About computational overhead. Reviewer qR3R requested detailed cost analysis. The authors provided tables for Llama3-8B/70B, quantifying trainable parameters (39.94KB for A³E vs. 0.54GB for ROME) and extra FLOPs (negligible vs. 16u+16v GFLOPs for 8B model inference). This confirmed A³E’s efficiency, addressing the concern.

4. About method clarity and fairness: (a) Distinguishing "incorrect preceding" and "knowledge sinking" (Reviewer cjhv): Authors provided contrasting examples (e.g., "fever, stomachache, loss of taste" = incorrect preceding; "fever → fever is a…" = knowledge sinking). (b) Baseline fairness (Reviewer dHwZ): Authors clarified baselines were adapted to CME (training edits independently, no latest-edit bias) and justified federated editing scenarios (GDPR-compliant edit sharing). Reviewer dHwZ acknowledged this resolved their "fair comparison" concern and raised their rating.

The authors proactively resolved reviewer concerns with concrete additions (e.g., notation tables, failure case analysis, running examples), strengthening the paper’s rigor and readability. All reviewers acknowledged the response.

**Decision Justification**

A³E fills a critical, understudied gap in model editing, with strong empirical support and practical utility. Unlike incremental improvements to single-edit methods, CME addresses a real-world limitation of LLMs (multifaceted query answering) that prior work ignored. This aligns with NeurIPS’ focus on impactful, real-world AI research. Meanwhile, the authors comprehensively addressed all major reviewer concerns: readability (notation tables, revised naming), evaluation (large models, classic datasets, failure cases), and computational cost (detailed FLOPs analysis). These additions transformed a borderline submission into a rigorous one.